# Efficient Active Imitation Learning with Random Network Distillation

**Emilien Biré[1]***, **Anthony Kobanda[2], Ludovic Denoyer[3], Rémy Portelas[2]**
[1]Centrale Supelec [2]Ubisoft La Forge [3]H Company
`remy.portelas@ubisoft.com`

## Abstract

Developing agents for complex and underspecified tasks, where no clear objective exists, remains challenging but offers many opportunities. This is especially true in video games, where simulated players (bots) need to play realistically, and there is no clear reward to evaluate them. While imitation learning has shown promise in such domains, these methods often fail when agents encounter out-of-distribution scenarios during deployment. Expanding the training dataset is a common solution, but it becomes impractical or costly when relying on human demonstrations. This article addresses active imitation learning, aiming to trigger expert intervention only when necessary, reducing the need for constant expert input along training. We introduce Random Network Distillation DAgger (RND-DAgger), a new active imitation learning method that limits expert querying by using a learned state-based out-of-distribution measure to trigger interventions. This approach avoids frequent expert-agent action comparisons, thus making the expert intervene only when it is useful. We evaluate RND-DAgger against traditional imitation learning and other active approaches in 3D video games (racing and third-person navigation) and in a robotic locomotion task and show that RND-DAgger surpasses previous methods by reducing expert queries. `https://sites.google.com/view/rnd-dagger`

## 1 Introduction

Imitation learning has increasingly become a favored approach for learning behaviors in complex environments, offering a compelling alternative to classical scripted behaviors implemented by domain specialists (Schaal, 1999; Hussein et al., 2017). It is particularly well suited in problems where there is not a clear performance measure (or reward). In video games, it is becoming more and more familiar to game developers that frequently address the problem of implementing bots in their games which must play in realistic ways (Harmer et al., 2018; Yadgaroff et al., 2024; Mao et al., 2024). Indeed, since the notion of realism is not well specified, it prevents the use of reinforcement learning-based approaches where a reward signal is mandatory. By observing and replicating human players, these bots are trained to execute complex strategies and actions that are both efficient and human-like.

Imitation learning usually proceeds in two steps: first, a dataset of behaviors is built by leveraging experts interacting with the dynamical system. Then, a statistical model (e.g. a neural network) is learned to imitate actions in that dataset, thus expecting this model to generalize to unseen state and to behave like the experts. Consider a use case where the goal is to train a driving policy in a video game context capable of controlling a car on a track (see figure 1). In this scenario, the state is defined by sensor values at time $t$, which can include information such as the position of the car, its speed, and raycasts. In this context, Imitation Learning involves manually controlling the car for many laps to create a learning dataset, training a policy on this dataset using traditional algorithms like Behavioral Cloning (Bain & Sommut, 1999), and expecting that the resulting bot will control the car correctly. However, this approach is known to be unreliable, particularly due to the risk of a shift in the state distribution between training and testing (Ross et al., 2011), known as the problem of covariate shift (Nair et al., 2019). When covariate shift occurs, the agent struggles to determine the correct action, leading to compounding errors and ultimately undermining its performance.

---

*Work done during an internship at Ubisoft La Forge.

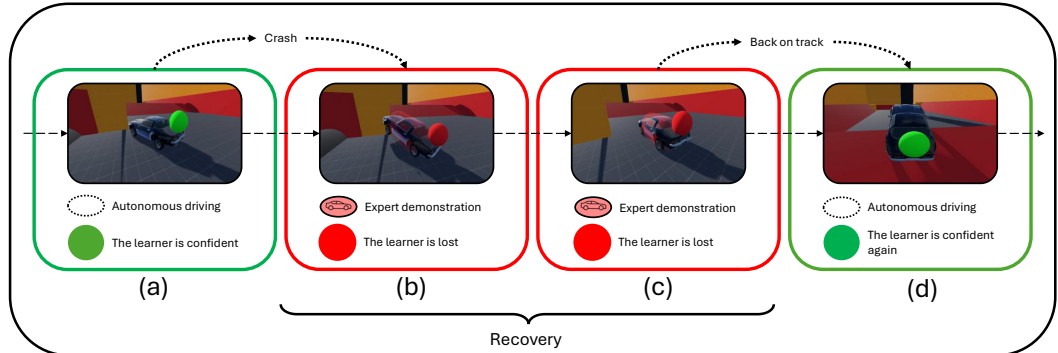

Figure 1: **RND-DAgger overview**. (a) The learner's policy controls the agent until (b) our Random Network Distillation-based out-of-distribution (OOD) measure is triggered. Then (c) the expert takes control until the OOD measure gets lower than the threshold for at least W steps. (d) Finally, the current policy takes control of the agent to continue the episode, and can trigger the expert again later.

Facing this issue, the effectiveness of imitation learning hinges on the availability of extensive datasets comprised of player behaviors, often necessitating thousands of expert traces to achieve reasonable performance and exhibit credible behaviors (Vinyals et al., 2019; Mao et al., 2024). This dependency on large volumes of data poses a challenge, particularly in scenarios where collecting such data is either impractical or costly.

In response to these limitations, some approaches involve human-in-the-loop training framework, often referred to as active imitation learning (Ross et al., 2011; Judah et al., 2012; Menda et al., 2019; Hoque et al., 2021). Instead of gathering very large amounts of demonstrations, these methods focus on the strategic incorporation of human feedback to selectively guide the learning process, thereby enhancing the efficiency of the training phase. By prioritizing the acquisition of relevant and impactful expert traces, these approaches seek to expedite the agent's learning curve without relying on expansive datasets, and swiftly improve performance. In the aforementioned video game example, this corresponds to collecting a first set of player traces to bootstrap the agent using classical imitation learning, and then to gather additional traces in particular race settings during training, i.e. requiring a player to control the car in particular and relevant situations.

To tackle these challenges, several recent models have been proposed to identify when the expert's intervention is necessary to correct the agent's suboptimal behavior (Zhang & Cho, 2017; Menda et al., 2019; Kelly et al., 2019; Hoque et al., 2021), with DAgger (Ross et al., 2011) being one of the most well-known approaches. The primary objective of these methods is to minimize the time required for human feedback by concentrating interventions on the most critical situations where the agent is likely to make errors. This allows for a more efficient use of expert time, ensuring that corrective feedback is provided only when it is essential. But, as explained in Section 2, many of these methods are not fully satisfying.

Despite these advancements, designing algorithms that effectively incorporate human feedback while reducing the overall expert effort remains an open research problem. Addressing this issue is the central focus of this article, as we aim to develop methods that optimize the interaction between the expert and the learning agent, thereby improving learning efficiency without overburdening the human supervisor.

Our main contributions are threefold: i) We propose a new method called RND-DAgger, a novel interactive imitation learning approach leveraging state-based out-of-distribution identification through random network distillation. ii) We perform a comparative analysis of RND-DAgger and existing methods on 3 tasks: a robotics scenario and two video-game environments. iii) Throughout these experiments, we demonstrate that RND-DAgger either outperforms or matches existing approaches in terms of final performance while significantly reducing expert burden.

**Algorithm 1** DAgger

**Require:** $K, \{\beta_i\}_{i \in [1,K]}, T, \pi_{exp}$
 1: Instantiate $D$ dataset of expert trajectories
 2: Instantiate $\pi_0$ BC policy trained on $D$
 3: $t \leftarrow 0$
 4: **for** $i = 0, \ldots, K$ DAgger iterations **do**
 5: $\quad$ $D_i \leftarrow \emptyset$
 6: $\quad$ **for** $T$ sampling steps **do**
 7: $\quad\quad$ $s_t \sim Env(a_{t-1})$ (Get a new state)
 8: $\quad\quad$ $a_t \leftarrow \beta_i \pi_{exp}(s_t) + (1 - \beta_i)\pi_i(s_t)$
 9: $\quad\quad$ $D_i \leftarrow D_i \cup \{(a_t, \pi_{exp}(s_t))\}$
10: $\quad\quad$ $t \leftarrow t + 1$
11: $\quad$ Push $D \leftarrow D \cup D_i$
12: $\quad$ Train $\pi_{i+1}$ on $D$ using BC
13: return $\pi_K$

Figure 2: **Above**, the DAgger algorithm where the expert action is added to the training set at each timestep, the agent being alternatively controlled by the current policy and the expert one. **On the right**, in Lazy/Ensemble DAgger, the agent is controlled by the expert policy only if a given measure is higher than a threshold $\lambda$. Only actions generated when the expert controls the agent are added to the training set.

**Algorithm 2** Lazy/Ensemble DAgger

**Require:** $K$, CONDITION, $T, \pi_{exp}$
 1: Instantiate $D$ dataset of expert trajectories
 2: Instantiate $\pi_0$ BC policy trained on $D$
 3: $nswitch \leftarrow 0$
 4: $t \leftarrow 0$
 5: $D_i \leftarrow \emptyset$
 6: **for** $i = 0, \ldots, K$ iterations **do**
 7: $\quad$ **while** $\#D_i < T$ **do**
 8: $\quad\quad$ $s_t \sim Env(a_{t-1})$ (Get a new state)
 9: $\quad\quad$ $C_t \leftarrow$ CONDITION$(\pi_{exp}, \pi_i, s_t)$
10: $\quad\quad$ **if** $C_t$ **then**
11: $\quad\quad\quad$ $a_t \leftarrow \pi_{exp}(s_t)$
12: $\quad\quad\quad$ $D_i \leftarrow D_i \cup \{(a_t, s_t)\}$
13: $\quad\quad\quad$ $t \leftarrow t + 1$
14: $\quad\quad\quad$ **if not** $C_{t-1}$ **then**
15: $\quad\quad\quad\quad$ $nswitch \leftarrow nswitch + 1$
16: $\quad\quad$ **else**
17: $\quad\quad\quad$ $a_t \leftarrow \pi_i(s_t)$
18: $\quad\quad$ $t \leftarrow t + 1$
19: $\quad$ $D \leftarrow D \cup D_i$
20: $\quad$ Train $\pi_{i+1}$ on $D$ using BC
21: return $\pi_K$

## 2 PRELIMINARIES AND RELATED MODELS

**Notations** Let us consider a Markov Decision Process described by a state space $\mathcal{S}$ and an action space $\mathcal{A}$. The dynamics of the process are defined by an unknown transition function $P(s_{t+1}|a_t, s_t)$, where $a_t \in \mathcal{A}$ and $s_t \in \mathcal{S}$. Note that we do not consider any reward in this setting since the task to solve is not explicit (e.g., driving a car); our objective is instead to imitate human behaviors that may be suboptimal and diverse (e.g., considering multiple players or experts). An episode $\tau$ consists of a sequence of states and actions, $\tau = (s_1, a_1, \ldots, s_T, a_T)$, where $T$ is the length of the episode. Given a policy $\pi(a_t|s_t)$, it is possible to sample an episode by sequentially executing the policy until reaching a stop criterion.

**Distributional shift:** Given a dataset of expert demonstrations $\mathcal{D} = (\tau_1, \ldots, \tau_n)$, a common approach to policy learning is Behavioral Cloning (BC) (Bain & Sammut, 1995; Ding et al., 2019a), a straightforward imitation learning algorithm. BC enables to learn policies by maximizing the log-likelihood of the expert actions within the dataset. However, learning a policy from a fixed dataset can result in the policy encountering out-of-distribution (OOD) states during inference, where it has not been trained to make accurate decisions (Ross et al., 2011). This phenomenon is referred to as distributional shift. This shift in the distribution can cause the policy to perform poorly, as it may not know how to handle unseen situations. For example, consider a driving policy in a racing game. If the dataset is composed only of demonstrations from expert drivers, the learned policy might struggle to recover from rare events such as a spin-out, since such situations are absent from the expert demonstrations. Therefore, the policy may fail to generalize effectively to these unseen states.

This effect can be mitigated by providing more diverse and comprehensive training sets, ensuring that the policy is exposed to a wider range of scenarios, including edge cases. However, this raises the challenge of how to efficiently collect such datasets while minimizing the burden on experts. Gathering enough data to cover all possible situations can be time-consuming and costly, especially if it relies heavily on expert demonstrations. Thus, the key question becomes how to strategically collect diverse and representative data in a way that maximizes coverage of the state space while minimizing the amount of effort required from experts.

| **Algorithm 3** Ensemble-DAgger's CONDITION | **Algorithm 4** Lazy-DAgger's CONDITION |
|---|---|
| **Require:** $\pi_{nov}$ a mixture of N policies, $\quad \pi_{exp}, s_t, \chi, \tau$ 
 1: $\sigma_{nov}^2 \leftarrow$ doubt of the mixture $\pi_{nov}$ 
 2: $m_t \leftarrow \|\pi_{exp}(s_t) - \overline{\pi_{nov}(s_t)}\|^2$ 
 3: **if** $\sigma_{nov}^2 > \chi$ **or** $m_t > \tau$ **then** 
 4: $\quad$ **return** TRUE 
 5: **else** 
 6: $\quad$ **return** FALSE | **Require:** $f, s_t, \beta_H, \beta_R$ 
 1: $m_t \leftarrow f(s_t)$ which estimates $\quad \|\pi_{exp}(s_t) - \pi_i(s_t)\|^2$ 
 2: **if** $m_t > \beta_H$ **then** 
 3: $\quad$ **return** TRUE 
 4: **else** 
 5: $\quad$ **if** $m_t < \beta_R$ **then** 
 6: $\quad\quad$ **return** FALSE 
 7: $\quad$ **else** 
 8: $\quad\quad$ **return** TRUE |

Figure 3: In Ensemble-DAgger, the policy is a mixture, and the measure is based on both a disagreement between the models in this mixture, and a disagreement with the expert action. The computations of these measure require the expert to provide action at each timestep of the process. In Lazy-DAgger, the measure is based on the disagreement between the current policy and the expert one, but to avoid the query of the expert action at each timestep, a classifier $f$ is trained to predict if the discrepancy measure $\|\pi_{exp}(s_t) - \pi_i(s_t)\|^2$ is above a given threshold $\beta_H$ or not.

**DAgger and variants:** To address this issue, one can utilize the DAgger algorithm from Ross et al. (2011) (Algorithm 1), which iteratively expands the training dataset through the intervention of an expert, represented by a **reference policy** denoted as $\pi_{exp}$. The reference policy serves two key roles: first, it determines the appropriate action to execute at each timestep based on a specified decision rule (line 7) – a mixture between the current policy and the expert one. Second, it helps augment the training set by adding new samples of states and corresponding expert actions, which are then used to improve the learned policy (line 11). At regular intervals, the policy is updated via imitation learning using the expanded dataset, and this process is repeated over multiple iterations. This method helps to mitigate the issue of distributional shift by continually refining the learned policy with fresh data that captures a broader range of states.

To better sample relevant states, few variants of the DAgger algorithm have been proposed. One of the most relevant is Ensemble-DAgger (Menda et al., 2019) which introduces a decision rule to decide if the agent is controlled by the current policy, or by the expert one. Moreover, pairs of state-actions are added to the learning dataset only when the action comes from the expert policy (see Figure 2 - right ). The decision rule is a discrepancy measure that computes the distance between the expert action and the policy action (combined with a disagreement measure between a mixture of experts) as shown in Algorithm 3. This approach is not realistic when considering human experts. Indeed, let us revisit our car driving example, where the goal is to learn an effective driving policy for a video game. Using the DAgger and Ensemble-DAgger approaches, the expert (i.e. the player) must continuously play the game during the active imitation learning process, while the car is only partially controlled by the current learned policy. This setup has two significant drawbacks: i) First, the player is placed in an unnatural setting, where they are expected to control a car that they do not fully manage. This can create a disorienting experience, as the player must constantly adapt to actions taken by the policy, disrupting the natural flow of gameplay. ii) Second, the player remains engaged even when the car is exhibiting good behavior, spending time providing feedback that may not significantly contribute to discovering a better policy. This leads to inefficiencies, as the expert's time is not always optimally utilized, especially in situations where the current policy already performs well.

## 3 RANDOM NETWORK DISTILLATION-BASED DAGGER

To address the aforementioned limitations of Menda et al. (2019), Lazy-DAgger (Hoque et al., 2021) proposes to involve the expert only during critical moments rather than requiring continuous control. Instead of relying on the expert to partially guide the agent throughout the learning process, this approach alternates between periods where the agent is completely controlled by the current policy and periods where it is fully controlled by the expert. During the phases controlled by the current policy, the expert's intervention is not required, thereby reducing the time the expert spends supervising the agent and making their interventions more targeted and efficient.

---

**Algorithm 5** RND-DAgger

**Require:** $K, \pi_{exp}, f_{targ}, f_{pred}, \lambda, W, T$
1: Instantiate $D$ dataset of expert trajectories
2: Instantiate $\pi_0$ BC policy trained on $D$
3: $nswitch \leftarrow 0$
4: $t \leftarrow 0$
5: $D_0 \leftarrow \emptyset$
6: **for** $i = 0, \ldots, K$ DAgger iterations **do**
7:     **while** $\#D_i < T$ **do**
8:         $s_t \sim Env(a_{t-1})$ (Get a new state)
9:         $m_t \leftarrow \|f_{targ}(s_t) - f_{pred}(s_t)\|^2$
10:         **if** $m_t > \lambda$ **or** $w < W$ **then**
11:             **if** $m_t \leq \lambda$ **then** (Minimal demo time)
12:                 $w \leftarrow w + 1$
13:             **else**
14:                 $w \leftarrow 0$
15:             $a_t \leftarrow \pi_{exp}(s_t)$
16:             $D_i \leftarrow D_i \cup \{(a_t, s_t)\}$
17:             **if** $m_{t-1} \leq \lambda$ **then**
18:                 $nswitch \leftarrow nswitch + 1$
19:         **else**
20:             $w \leftarrow 0$
21:             $a_t \leftarrow \pi_i(s_t)$
22:         $t \leftarrow t + 1$
23:     Push $D \leftarrow D \cup D_i$
24:     Train $\pi_{i+1}$ on $D$ using BC
25:     Train $f_{pred}$ on $D$ to predict $f_{targ}$
26: **return** $\pi_K$

---

Figure 4: The **RND-DAgger algorithm** proceeds in $K$ iterations. First a policy is trained on a small expert dataset (line 2). Then at each iteration (line 6), it constructs a new dataset $D_i$ with $T$ state-action pairs from the expert (line 7). At each timestep, the agent is by default controlled by the current policy (line 23). The OOD measure is computed by using the prediction network $f_{pred}$ and the target network $f_{targ}$ (line 9). If this measure is greater than a threshold (line 10), the expert takes over the policy to control the agent (line 16) and to add samples to the dataset (line 17). The number of context switches is the number of times the expert has taken control (line 19). At the end of the iteration, the built dataset is aggregated with the current learning set (line 27) and the policy is retrained (line 28). In addition, the $f_{pred}$ network is also updated (line 29) to allow the future detection of OOD states. The $w$ counter is used to ensure that the algorithm waits at least $W$ timesteps below the threshold before taking control back of the agent.

Lazy-DAgger replaces the traditional action-based discrepancy measure between expert and policy actions with a classifier-based approximation — see Figure 4. This approach allows the algorithm to predict when expert intervention is necessary without requiring an expert action at every timestep, thereby significantly reducing the expert burden compared to methods like Ensemble-DAgger. However, this method has a notable drawback: similar to Ensemble-DAgger, it relies on comparing the actions of the expert and the current policy for a given state. While this approach works well when the expert is optimal and acts deterministically, it becomes problematic when dealing with humans or imperfect experts. Human experts often exhibit variability and may choose different actions for the same state depending on context or personal preference. This results in a noisy and unreliable measure of discrepancy. To address this issue, one could rely on a state-based discrepancy measure that would focus on meaningful divergences compared to an action-based one.

We propose the RND-DAgger algorithm, which builds upon Random Network Distillation (RND) (Burda et al., 2019). The core assumption of RND-DAgger is that expert feedback is only necessary when the agent encounters out-of-distribution (OOD) states—states that are not well represented in the training set and where the policy is more likely to fail. This is crucial because when an agent operates in OOD states, it faces a higher risk of taking suboptimal actions that could hinder learning or lead to unsafe outcomes. Once the agent returns to familiar, in-distribution states, the expert's feedback is no longer required, and the intervention is ended. Our method differs fundamentally from existing approaches like Ensemble-DAgger and Lazy-DAgger, which rely on action-based discrepancy measures to determine when to intervene. As such, our approach remains robust to variations in expert behavior, ensuring that interventions only occur when the agent genuinely faces unfamiliar and potentially risky states.

To further improve the stability of expert interventions, we introduce a mechanism called minimal demonstration time. This concept addresses the issue of overly frequent and brief expert interventions, which can interrupt learning and increase the cognitive load on the expert. Minimal demonstration time defines a lower bound on the duration for which the expert must maintain control once they start an intervention. The key idea is that the expert should provide a sufficient number of consecutive corrective actions to guide the agent back to a stable state, instead of immediately handing control

back to the policy after a single correction. We describe these two components below. The detailed algorithm is provided in Algorithm 4 with a complete explanation of the different steps.

**Random Network Distillation**    To measure if a state is OOD, RND-DAgger relies on the Random Network Distillation technique, which is a classical approach developed initially for the problem of exploration in Reinforcement Learning to detect new states to explore (Burda et al., 2019). The principle of RND is to use a randomly initialized neural network as a fixed target $f_{targ}$ and train a second neural network $f_{pred}$ (the predictor) to approximate the output of the target network. As the predictor network improves over time, the error between the predictor and the target network decreases for familiar states (in-distribution), but remains high for unseen or out-of-distribution states. This prediction error thus serves as a measure of novelty, allowing the agent to recognize when it has encountered a new or unfamiliar state. In RND-DAgger, the measure is used to decide when to trigger the expert, and when to trigger back the current policy, alternating phases where the expert is controlling the agent and when the agent is controlled by the current policy. Note that, contrary to the described baselines, this measure does not involve the expert action, thus avoiding the need to have an expert acting at each timestep.

**Minimal demonstration time**    Relying exclusively on the OOD measure can lead to a behavior where the expert is asked too often for short demonstrations (see appendix Figure C.2). To account for this limitation, we introduce the notion of minimal expert time (MET). It is defined as $W$ the number of consecutive "in-distribution" frames required to switch control back from the expert to the autonomous policy (i.e. the number of frames where $m_t < \|f_{targ}(s_t) - f_{pred}(s_t)\|^2$). This mechanism ensures that the expert will provide a minimal amount of information at each intervention, resulting in fewer, but longer, demonstration sequences.

## 4    EXPERIMENTS

The primary objectives of our experimental evaluation are twofold: first, to assess the efficiency of our approach in discovering an effective policy, and second, to evaluate its ability to reduce the expert's burden.

**Environments:**    Our first environment is **HalfCheetah** which is a classical reinforcement learning environment[1] where the objective is to learn a running strategy for the agent. The goal of the agent is to locomote as fast as possible. The HalfCheetah is controlled by applying motor torques, and the agent manipulates a 6-degree-of-freedom (6-DOF) motor joint vector. The observation space consists of 18 values, which include the position and velocity of the agent's body, the angles and angular velocities of its six joints, as well as the angle of the center of mass. We also propose and open-source two new environments developed for video game research . **RaceCar** (see Figure 5) features a physics-based car controller that must complete a single lap on a given track. After each lap, the car is reset to a random position on the starting line. The track presents several challenges, including speed bumps, a ramp in front of a pillar, and sharp 90-degree turns, making the optimal driving behavior non-trivial. Additionally, crossing the ramp and red sloped walls provides a speed boost, adding complexity to the strategy. The car is controlled using four discrete actions: forward, backward, left, and right. The observation space consists of 22 dimensions, including the car's 3D position, linear velocity (3D), angular velocity (3D), rotation (encoded as the cosine and sine of the angle), and data from seven raycasts to detect obstacles. Finally, the **3D Maze** environment allows us to study our strategy in goal-conditioned navigation scenarios. A classical human-like character is spawned at a random location within the maze and tasked with reaching a randomly assigned goal. The agent is controlled using four movement actions: walk forward, walk backward, strafe left, and strafe right, as well as a one-dimensional rotation action to change its facing direction.The observation space consists of the agent's absolute position (3D), the goal's position (3D), and 131 raycasts, which provide a low-resolution depth map of the surroundings. Since the objective in this environment is to navigate to any goal location, we replace the traditional Behavioral Cloning (BC) approach with a Goal-Conditioned Behavioral Cloning (GCBC) approach (Ding et al., 2019b) as the base learning algorithm. This adaptation ensures that the agent learns not just to imitate expert

---

[1]https://github.com/araffin/pybullet_envs_gymnasium

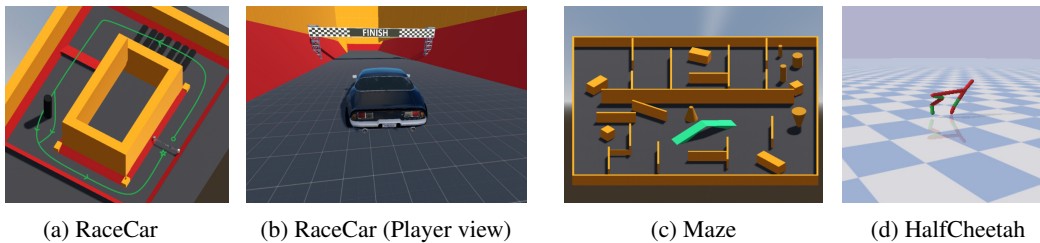

(a) RaceCar          (b) RaceCar (Player view)          (c) Maze          (d) HalfCheetah

Figure 5: Illustration of the three different environments used in our experiments. The objective is to learn a good driving, navigating, and walking policy while minimizing expert interventions.

trajectories, but also to generalize its navigation strategies based on varying goal positions within the maze.

**Baselines:**    We compare our approach RND-DAgger to multiple approaches from the active imitation learning literature: **DAgger** Ross et al. (2011), a classical approach which queries the supervisor for an action in every state that the learner visits. **Ensemble-DAgger** Menda et al. (2019), which propose an automated approach to measure OOD and trigger expert demonstration. Ensemble-DAgger relies on two metrics: A disagreement measure defined as the correlation between an ensemble of policies, and a discrepancy measure computing the difference between the bot action and the expert action at every step. **Lazy-DAgger** uses the same discrepancy measure as Ensemble-DAgger and reduces bot-expert context switching by using an additional threshold parameter to define a hysteresis band. Note that, we rely on a simplified version proposed in (Hoque et al., 2021) that has been shown to be more efficient and generates less context switches than the original version. This variant discards the classifier for the benefit of the true discrepancy measure. **Human-Gated DAgger (HG-DAgger)** from Kelly et al. (2019) involves a human supervisor monitoring the agent's behavior in real-time and deciding when to take control. While similar in essence to our proposed RND-DAgger, HG-DAgger relies entirely on human judgment to identify when intervention is necessary, rather than using an automatic OOD measure. This requires the expert to continuously observe the agent's actions at every timestep, leading to a higher cognitive load and less efficient use of expert time. Finally, we also consider **Behavioral Cloning (BC)** as the classical baseline, which does not include any active learning mechanism.

The set of hyperparameters used for the different approaches is described in Appendix A. For each method, a first training set is collected by the expert, without active imitation learning techniques. Then the different algorithms are executed to collect additional examples and to update the policy.

**Metrics:**    To assess the performance of the methods considered, we rely on several key metrics: i) The *Task Performance* measures whether the learned agent is successfully solving the environment. For the **Race Car** and **Maze** environments, we use the success rate (i.e., whether the agent completes a lap or reaches the goal). For **HalfCheetah**, we track the cumulative episode reward, which reflects the overall effectiveness of the agent's locomotion strategy. ii) The *Dataset Size* indicates the number of expert actions added to the training dataset, providing insight into how much expert information is being utilized to train the agent. A smaller dataset size suggests the agent is learning efficiently with fewer expert interventions, while a larger dataset implies greater reliance on expert input. iii) The *Context Switches* which corresponds to the $nswitch$ variable in Algorithms 1 and $\omega$ measures the number of contiguous periods during which the expert is actually controlling the agent. This helps quantify how often the expert is involved in directly providing relevant training samples. For RND-DAgger, this measure accurately reflects the expert's involvement, as the agent is fully controlled by the policy between switches. The metrics are averaged over 8 seeded runs for each method.

## 4.1 RESULTS

**Oracle-based Performance:**    To conduct extensive experiments, we replace the human expert with a predefined oracle policy. This substitution allows us to run multiple trials efficiently without depending on human participants, which would be impractical in terms of both time and resources (experiments using human experts are reported separately later in the section). For the Race Car

| | Task Performance | | | # Context Switch | | |
|---|---|---|---|---|---|---|
| Env
Method | RC | HC | Maze | RC | HC | Maze |
| BC | $0.883 \pm 0.029$ | $2455 \pm 424$ | $0.367 \pm 0.043$ | - | - | - |
| DAgger | $0.940 \pm 0.026$ | $2343 \pm 209$ | $0.450 \pm 0.074$ | - | - | - |
| Lazy-DAgger | $0.939 \pm 0.025$ | $2314 \pm 278$ | $0.575 \pm 0.101$ | $3437 \pm 89$ | $\mathbf{312} \pm 45$ | $\underline{1238} \pm 79$ |
| Ensemble-DAgger | $\mathbf{0.952} \pm 0.018$ | $\underline{2489} \pm 108$ | $\underline{0.626} \pm 0.045$ | $2785 \pm 41$ | $1452 \pm 134$ | $2871 \pm 94$ |
| RND-DAgger | $\underline{0.944} \pm 0.014$ | $\mathbf{2490} \pm 160$ | $\mathbf{0.717} \pm 0.018$ | $\mathbf{368} \pm 11$ | $\underline{708} \pm 130$ | $\mathbf{1214} \pm 25$ |

Table 1: **Overall performance**: This table illustrates the performance of the different algorithms at the end of the K iterations. It also provides the value of the nswitch variable that accounts for the number of expert interventions. **Best**, Second best

environment, the oracle is learned through a separate active imitation learning process with a large interaction budget, ensuring the agent has access to high-quality demonstrations. In the 3D Maze environment, the oracle is a NavMesh-based agent, which programatically solves any navigation scenario, making it easy to generate trajectory data. For HalfCheetah, we use an open-source reinforcement learning agent (Kuznetsov et al., 2020) from the RL Zoo repository (Raffin, 2020) as the oracle, providing an optimal policy for locomotion tasks. This approach ensures consistency and scalability across experiments.

Table 1 and Figure 6 present the performance metrics across the evaluated environments. To ensure a fair comparison, we report results based on the hyperparameter settings that achieve the highest final performance for each model.

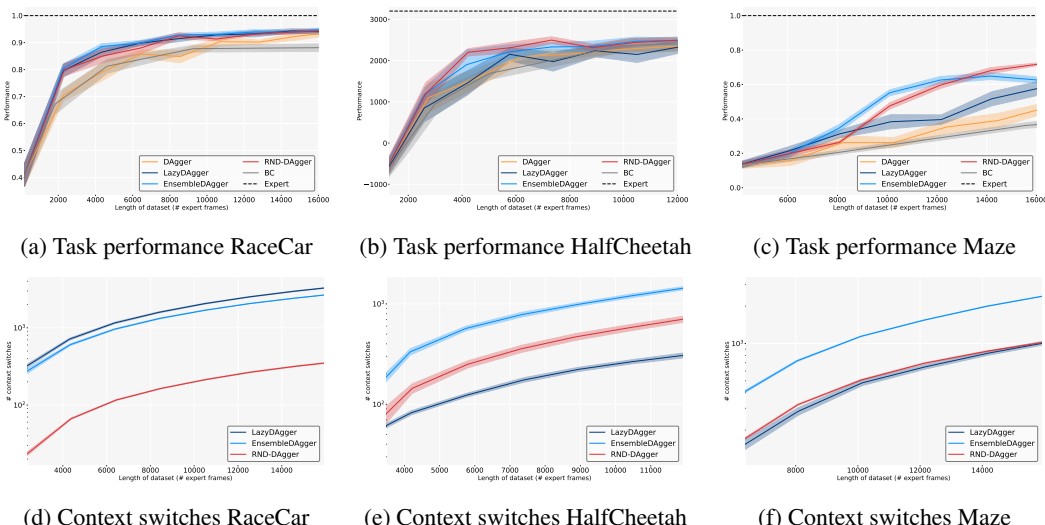

(a) Task performance RaceCar    (b) Task performance HalfCheetah    (c) Task performance Maze

(d) Context switches RaceCar    (e) Context switches HalfCheetah    (f) Context switches Maze

Figure 6: Performance and context switches for the different environments. The X-axis corresponds to the size of the training set $D$ that increases at each iteration of the algorithms.

Our method, RND-DAgger, demonstrates competitive performance relative to Ensemble-DAgger and Lazy-DAgger, indicating that it effectively learns robust policies across different environments. For example, in the HalfCheetah environment, RND-DAgger achieves a cumulative reward of 2490, compared to 2489 for Ensemble-DAgger and 2314 for Lazy-DAgger. These results highlight that RND-DAgger is capable of discovering efficient policies, matching or surpassing the baselines in terms of overall task success. When analyzing performance relative to the size of the training set, RND-DAgger outperforms the other methods in the early stages of the active imitation learning process (Fig. 6b). This suggests that RND-DAgger is more effective at focusing on critical states where expert guidance is most needed, thereby gathering valuable feedback more efficiently. By concentrating on key interventions, RND-DAgger accelerates policy improvement and requires fewer training samples to achieve comparable or superior results, making it especially advantageous in scenarios with limited expert availability. In summary, the experimental results demonstrate that RND-DAgger not only achieves strong final performance but also exhibits a more sample-efficient

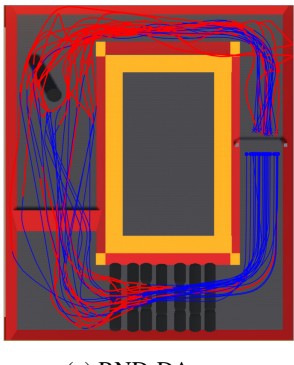 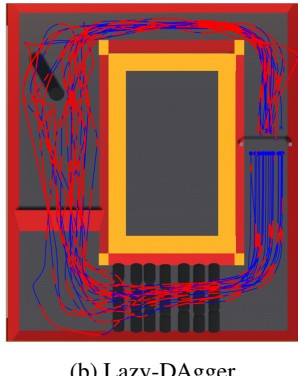 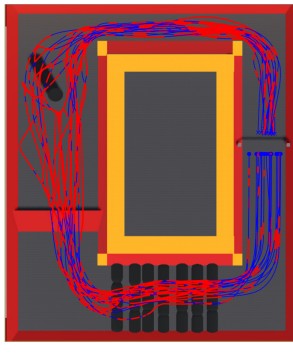

|(a) RND-DAgger | (b) Lazy-DAgger | (c) Ensemble-DAgger |

Figure 7: Visualization of agent's trajectories (blue) and expert intervention trajectories (red) in RaceCar. RND-DAgger predominantly requests expert intervention in challenging sections of the track, requiring fewer expert interventions overall compared to baseline methods.

learning curve compared to existing methods (Fig. 6a, 6b and 6c), validating its capability to optimize expert interventions and rapidly improve policy quality.

When examining the number of context switches between the different methods, RND-DAgger consistently results in significantly fewer context switches compared to Ensemble-DAgger across all environments. This indicates that RND-DAgger is more stable and requires fewer handovers between the expert and the policy, reducing the burden on the expert. When compared to Lazy-DAgger, RND-DAgger shows mixed results. In the RaceCar environment, RND-DAgger generates fewer context switches (Fig. 6d), highlighting its efficiency in minimizing expert interventions. However, in the HalfCheetah and Maze environments, RND-DAgger produces a similar number of switches (Fig. 6e and 6f). This similarity in HalfCheetah can be attributed to the use of an oracle expert policy, which performs consistently without generating diverse actions. Under these circumstances, the discrepancy measure used by Lazy-DAgger, which relies on comparing expert and policy actions, remains effective for detecting when to switch. In the Maze environment, despite having a comparable number of context switches to Lazy-DAgger (Fig. 6f), RND-DAgger achieves a higher final performance (Fig. 6c).This suggests that RND-DAgger intervention criterion leads to more informative expert interventions and is better able to leverage the expert. Thus, in scenarios like Maze, where achieving high task performance is critical, RND-DAgger is the better choice due to its ability to drive the agent towards optimal behavior.

Overall, these findings indicate that RND-DAgger strikes a favorable balance between reducing context switches and achieving strong policy performance, making it a more efficient and reliable choice.

**Qualitative study:** To gain a deeper understanding of when our algorithm requests expert supervision, Figure 7 visualizes the context switches between the current policy and the expert (additional figures are provided in Appendix B). As seen in the figure, both Ensemble-DAgger and Lazy-DAgger query the expert significantly more frequently, leading to a higher number of context switches compared to RND-DAgger. The visualization also reveals that RND-DAgger identifies and focuses on critical areas of the track that are challenging for the agent during early learning stages. Specifically, it requests expert intervention primarily in the problematic zones, such as the bottom section of the track and the obstacle immediately following the speeding ramp (top-left corner of the track). These are regions where the bot initially struggles, making precise interventions crucial for improving performance. This observation indicates that RND-DAgger not only minimizes the number of expert queries but also targets the most relevant segments of the environment where guidance is essential. By focusing on challenging states rather than spreading expert effort across all states, RND-DAgger makes more strategic use of expert supervision, resulting in fewer yet more impactful interventions.

**Human Expert-based performance:** We conducted a series of experiments (Figure 8) using a real human expert instead of an oracle policy. In this setup, collecting expert actions while the agent was being controlled by its own policy (rather than having the expert take direct control) felt unintuitive for the human participants, making it challenging to apply traditional baselines effectively.

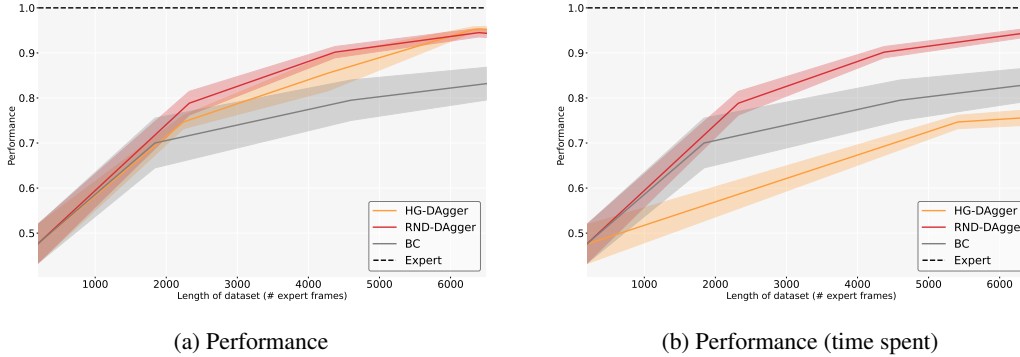

(a) Performance                          (b) Performance (time spent)

Figure 8: Task performance on RaceCar with a human expert. (a) RND-DAgger achieves similar task performance to HG-DAgger with a comparable number of demonstrations. (b) However, by reducing the need for constant expert monitoring, RND-DAgger significantly lowers the total expert time required.

As such we focused our comparison on HG-DAgger, which allows the human expert to decide when to intervene and take control from the agent.

Our results show that RND-DAgger performs comparably to HG-DAgger, but with a significant advantage: it does not require the human expert to continuously monitor the agent's behavior. Instead, RND-DAgger automates the decision of when to request expert input based on state novelty, reducing the cognitive load on the expert. Figure 8b demonstrates the relationship between policy performance and the actual time the expert spends observing the screen. In RND-DAgger, the agent can run at accelerated speeds without the need for constant supervision, as the expert only needs to intervene at specific moments rather than watching the agent's entire trajectory.

**Additional experiments** Appendix C presents ablation studies of RND-DAgger, including an analysis of the impact of the Minimal Expert Time mechanism (showing how it reduces expert context switches). Appendix D focuses on studying the impact of using Ensemble-DAgger compared to RND-DAgger regarding expert time spent (the metric used in Figure 8b), showing how RND-DAgger better minimizes expert burden.

## 5  CONCLUSION AND LIMITATIONS

In this work, we presented RND-DAgger, a novel approach to active imitation learning that efficiently minimizes the need for expert interventions by leveraging a state-based measure derived from Random Network Distillation (RND). Unlike traditional methods that rely on action-based discrepancies to detect when to seek expert guidance, RND-DAgger focuses on identifying out-of-distribution states where the agent is most at risk of making errors. This allows our method to selectively request expert feedback only when it is truly necessary, thereby reducing the number of context switches and optimizing the allocation of expert time. Overall, our method provides a step forward in developing more practical and human-efficient imitation learning algorithms, making it a valuable tool for training autonomous agents in complex environments.

**Limitations** RND-DAgger requires the expert to immediately take control of the agent, which may be impossible or dangerous in complex real-time environments. Predictive approaches to alert the expert before intervention is an interesting area of improvement. Studying how to scale RND-DAgger to more complex environments such as Partially observable MDPs – e.g. pixel-based environments – would be a natural extension of our work. In such scenarios, noisy-TV problems i.e. environments featuring distracting random or diverse observations, bring important new challenges, which may be addressed through representation learning or explicit high-entropy filtering. Lastly, incorporating richer forms of expert feedback is another valuable future line of work.

## 6 ACKNOWLEDGEMENTS

This work was granted access to the HPC resources of IDRIS under the allocation 2024-AD011015218 made by GENCI.

## 7 REPRODUCIBILITY STATEMENT

To ensure the reproducibility of our work, we provide detailed pseudo-code in section 2 and section 3. A comprehensive open-source codebase, including all environments, datasets, oracle model checkpoints, active learning algorithms, and a detailed guide on how to reproduce our experiments and results is available at `https://sites.google.com/view/rnd-dagger`.

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

# A EXPERIMENTAL DETAILS

**Dataset** Figure 9a and 9b showcase examples of generated trajectories for the RaceCar and Maze environments.

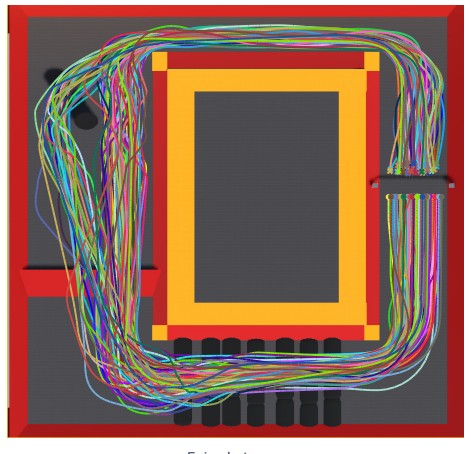

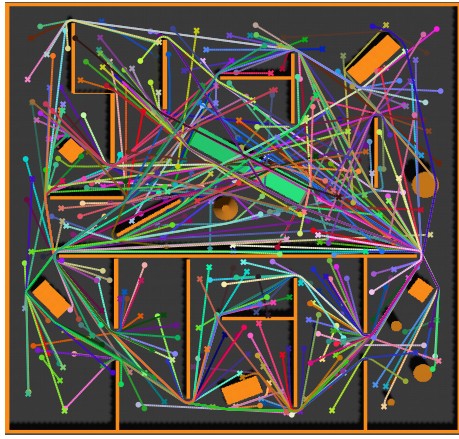

(a) RaceCar dataset examples        (b) Navigation dataset examples

Figure 9: Dataset examples: the trajectories of the expert from which we bootstrapped to train our initial BC policies and out BC baselines.

**Hyperparameters** For each decision rule, several key hyperparameters had to be tuned:

- **DAgger**
  - The probability $\beta$ of a frame to be controlled by the bot. The probability is decreased at each DAgger epoch by $\beta_i \leftarrow \beta_0^{i-1}$
- **RND-DAgger**
  - Threshold $\lambda$ of OOD detection
  - The historic context length, that is the number of frames in the past we take along with the current state to detect its OOD nature
  - The Minimal Expert Time $W$
  - The size of the random network, represented by the number of layers and neurons per layer the predictor and target networks have.
- **Ensemble-DAgger**
  - Threshold $\tau$ for discrepancy measure
  - Threshold $\chi$ for doubt measure
  - The number of models $N$
- **Lazy-DAgger**
  - Threshold $\beta_H$ for discrepancy measure
  - Threshold $\beta_R$ for the backward controlled loop (i.e. the criterion to switch back from expert to autonomous)

**Computing thresholds** For all the methods that necessitate at least one threshold (RND-DAgger, Lazy-DAgger, Ensemble-DAgger), we computed them following the following methodology: the threshold was set to be a positive factor of the mean measure on the training set. In other words, before each new sessions, the measure was ran over the training dataset, and the threshold ($\chi$ and $\tau$ for Ensemble-DAgger, $\beta_H$ for Lazy-DAgger and $\lambda$ for RND-DAgger) were calculated as :

$$\lambda \leftarrow \overline{\text{MEASURE}(\{(a_t, s_t) \in D_{train}\})} \times L \qquad (1)$$

Table 2: Hyperparameter search

(a) Ensemble-DAgger

| Hyperparameter | Values | RaceCar | HalfCheetah | Maze |
|---|---|---|---|---|
| $\chi$ factor | $[0, 1, 1.5, 2, 3, 4]$ | 1.5 | 0 | 0.5 |
| $\tau$ factor | $[0, 1, 1.5, 2, 3, 4]$ | 1.5 | 1.5 | 3 |
| N | $[2, 3, 5]$ | 5 | 5 | 3 |

(b) RND-DAgger

| Hyperparameter | Values | RaceCar | HalfCheetah | Maze |
|---|---|---|---|---|
| $\lambda$ factor | $[1, 2, 3, 4]$ | 2 | 2 | 2 |
| Hidden size | $[32, 128]$ | 32 | 128 | 32 |
| Number of layers | $[0, 1, 2]$ | 0 | 2 | 0 |
| Historic context length | $[0, 2, 5, 10, 15]$ | 10 | 0 | 2 |
| $W$ | $[1, 2, 5, 10, 15, 30, 50]$ | 30 | 5 | 5 |

(c) Lazy-DAgger

| Hyperparameter | Values | RaceCar | HalfCheetah | Maze |
|---|---|---|---|---|
| $\beta_H$ factor | $[0, 1, 2, 3, 4]$ | 1.5 | 0 | 2 |
| $\beta_R$ divider | $[1, 1.5, 2, 3, 4]$ | 2 | 1.5 | 2.5 |

(d) DAgger

| Hyperparameter | Values | RaceCar | HalfCheetah | Maze |
|---|---|---|---|---|
| $\beta$ | $[0.2, 0.5, 0.6, 0.7, 0.8, 0.9, 0.95]$ | 0.5 | 0.7 | 0.8 |

With L being a positive factor, a hyperparameter to be tuned. For the second threshold of Lazy-DAgger, we set it to be a factor of the first one: $\beta_R = \beta_H/L^*$, with $L^*$ being the hyperparameter tuned. That's a different method compared to other methods, such as Zhang & Cho (2017) or Hoque et al. (2021) who set their threshold so that approximately 20% of the initial dataset is unsafe, or Ensemble-DAgger Menda et al. (2019) that directly grid searched the value. The Table 2 summarizes the values used for our grid search.

# B    INTERACTIVE SESSION VISUALIZATIONS

**Typical interactive situations**    In Figure 11, one can understand better how, in the RaceCar environment, an oracle (and by extend a human taken as an expert) would interact with the learner. In (a), the current learned policy has control of the car, and the OOD measure (green circle) is below the threshold: the learner is confident. In (b), the car crashes into a wall, so the measure (red circle) queries the expert to take control (transparent red capsule). From (c) to (e), the expert demonstrates how to get back on track, until (f) the measure falls below the threshold again (green circle) for at least $W$ consecutive frames (see Section 3 for further explanations)

**Qualitative study**    On Figure 10, we reported the context switches and full trajectories on both the RaceCar and Maze environments. The results are taken from the same session and the same seed to better compare the figures. From (a) to (b), it is clear that Ensemble-DAgger exhibits the most context switches compared to Lazy-DAgger and RND-DAgger, which show comparable counts. However, the context switches in Lazy-DAgger are more concentrated in specific areas, whereas those in RND-DAgger are more evenly distributed. Then, from (d) to (f) we can see the corresponding episodes with expert demonstrations (in red) added to the dataset during that session. We can see that the expert demonstrations are either way longer (the whole episode) or way shorter (only a few frames) for Lazy-DAgger compared to our method, providing more consistency throughout the map. The difference is clearer on the RaceCar environment. Indeed, on both context switch figures, we see that Lazy-DAgger and Ensemble-DAgger are comparable, and that the queries for RND-DAgger are concentrated around only two zones that generated most of the failure cases.

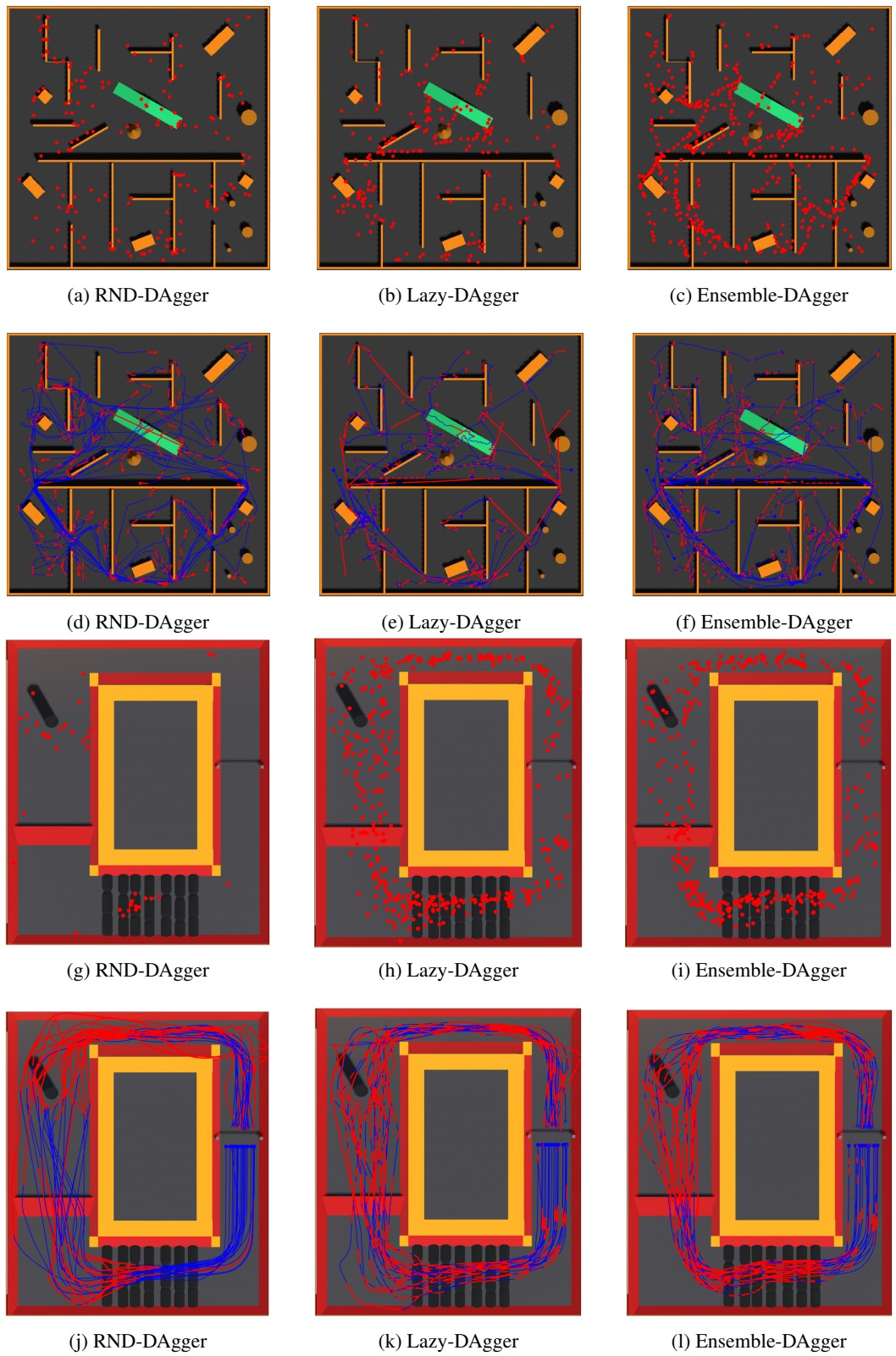

(a) RND-DAgger     (b) Lazy-DAgger     (c) Ensemble-DAgger

(d) RND-DAgger     (e) Lazy-DAgger     (f) Ensemble-DAgger

(g) RND-DAgger     (h) Lazy-DAgger     (i) Ensemble-DAgger

(j) RND-DAgger     (k) Lazy-DAgger     (l) Ensemble-DAgger

Figure 10: We conduct a qualitative analysis of the the different query methods. (a)-(d) and (g)-(i): Each dot represents a context switch : a transition from autonomous to expert control. (e)-(h) and (j)-(l): In red, the expert demonstrations, in blue the novice segments.

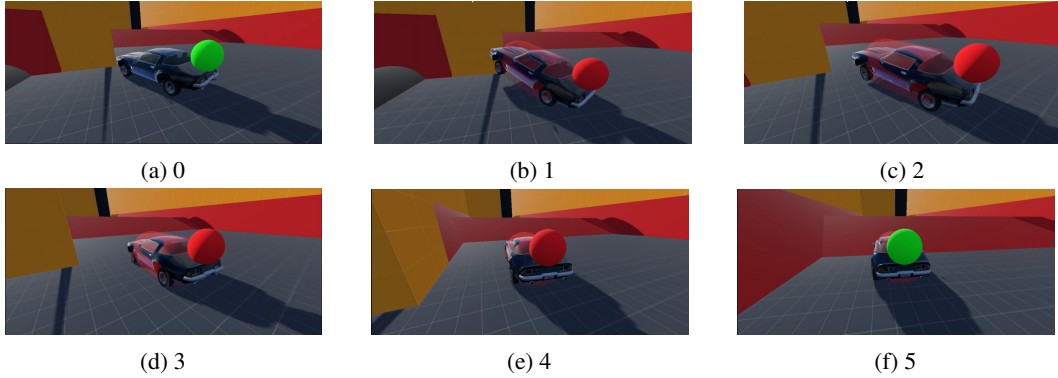

(a) 0          (b) 1          (c) 2

(d) 3          (e) 4          (f) 5

Figure 11: (a) The agent has control. (b) The OOD measure is triggered (RND), the expert is given control. (c)-(e) The expert demonstrates how to recover. (f) The state is in distribution, the expert keeps the control until the Minimal Expert Time is reached.

## C  ABLATION STUDY

### C.1  RND-DAGGER ON HALFCHEETAH

In this ablation study, we evaluate RND-DAgger, focusing on the additive advantages of three key enhancements: a **Minimal Expert Time (MET)** of size $W$, **historical context**, and the **architecture** of the random networks (see Section 2 for insights on the values tested). The addition of a Minimal Expert Time should increase the quality of the frames by letting the expert finish its demonstration and making sure the learner is fully recovered before letting it the control again. The historical context enriches decision-making by incorporating temporal information from past actions and concatenating them to the current state. This particularly helps in tasks where there is a strong relation between the states. For example, a state where the car is close to a wall is not necessarily a bad state, unless the car is going towards it for several frames in a row. However, a historical context chosen too big is detrimental, because it can overshadow useful dimensions of the sate, and thus preventing the predictor network to seize useful information.

Results of that ablation are reported in Table 3 and Figure 12. We note that the Minimal Expert Time $W$ helps reducing the context switching, without impacting that much the performance at the end of the sessions of RND-DAgger. In other words, it helps decreasing the expert burden, without impacting the task performance.

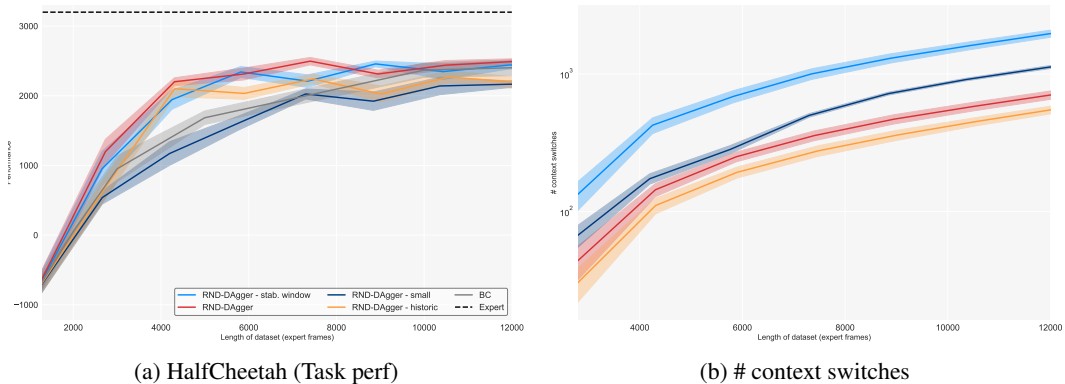

(a) HalfCheetah (Task perf)          (b) # context switches

Figure 12: Ablation study on our method. RND-DAgger compared to: (light blue) RND w/o MET, (orange) bigger historical context, (dark blue) smaller architecture for the random networks.

|  | Task Performance | # Context Switch |
|---|---|---|
| BC | $2455 \pm 424$ | - |
| RND-DAgger | $\mathbf{2490} \pm 160$ | $708 \pm 130$ |
| RND-DAgger - MET | $\mathbf{2432} \pm 175$ | $1985 \pm 215$ |
| RND-DAgger - small | $2167 \pm 140$ | $1136 \pm 76$ |
| RND-DAgger - historic | $2172 \pm 290$ | $\underline{554} \pm 90$ |
| Lazy-DAgger | $\underline{2314} \pm 278$ | $\mathbf{312} \pm 45$ |

Table 3: Performance and context switches at the end of the sessions. We report the results of the key ablations on RND-DAgger, along with the results of Lazy-DAgger for further comparisons

## C.2 MINIMAL EXPERT TIME FOCUS

**Ablation study on all the methods**    We further investigated the impact of the Minimal Expert Time (MET) on our method and on our baselines, across all the tasks. We reported the results of that study in Figure 13 and Table 4. In particular, we can see that LazyDAgger + MET isn't appropriate in HalfCheetah. The combination of the Lazy mechanism (introduction of a second threshold to create a hysteresis band) and the MET forces the expert to always have control over the agent, which explains the extremely low value for the context switches (which happens only once per episode) and the poor performance learning curve (Fig. 13b). The same remark holds for Ensemble-DAgger on RaceCar, but in this case, this behavior doesn't seem to impact the task performance of the method (Fig. 13a).

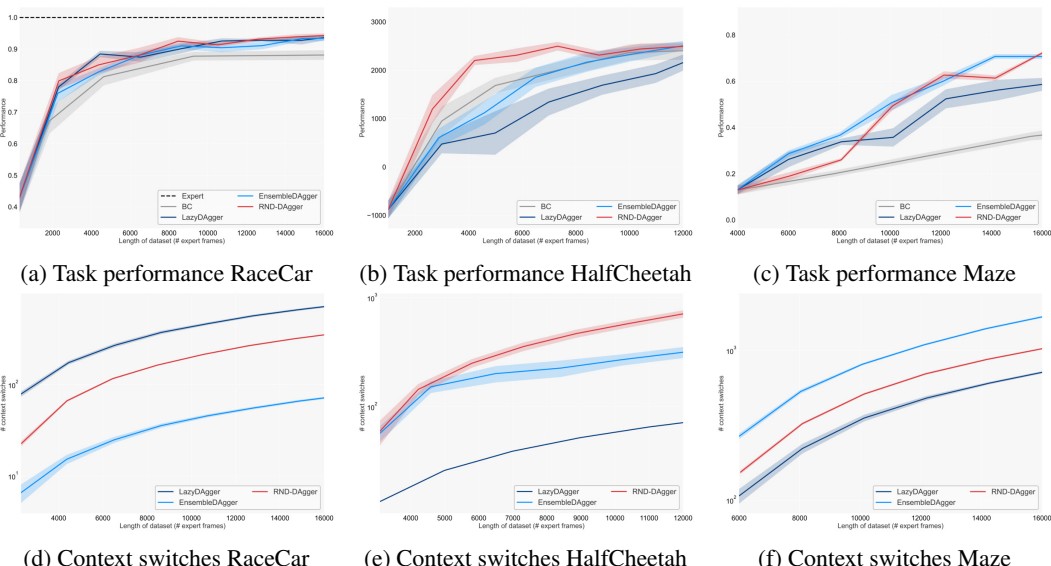

(a) Task performance RaceCar    (b) Task performance HalfCheetah    (c) Task performance Maze

(d) Context switches RaceCar    (e) Context switches HalfCheetah    (f) Context switches Maze

Figure 13: Results for the ablation study with focus on the addition of the Minimal Expert Time (MET) on the baselines Ensemble-DAgger and LazyDAgger. RND-DAgger remains the same as in Table 1

**Qualitative interpretation**    On Figure 14 we report the context switches on the RaceCar environment, at the second DAgger epoch of a given seed with and without a MET. We can clearly see that the number of context switches increases and a third critical zone appeared at the top right of the map for RND-DAgger (Fig. 14a). We interpret that the bot is fairly confident in the top straight line of the course, but the last turn is difficult to manage when the velocity of the car is too high, inducing a query to the expert. There are also more context switches at the entrance of that straight line, because the learner has to be precise to pass without touching the walls or the speed ramps on the sides. We conclude that the MET increases the quality of the demonstrations without impacting the quality of the measure, that still grasps the relevant critical areas of the track.

| | Task Performance | | | # Context Switch | | |
|---|---|---|---|---|---|---|
| | RC | HC | Maze | RC | HC | Maze |
| Lazy-DAgger + MET | $0.943 \pm 0.011$ | $2374 \pm 325$ | $0.575 \pm 0.101$ | $759 \pm 39$ | $78 \pm 0.0$ | $876 \pm 71$ |
| Lazy-DAgger | $0.939 \pm 0.025$ | $2314 \pm 278$ | $0.590 \pm 0.057$ | $3437 \pm 89$ | $312 \pm 45$ | $1238 \pm 79$ |
| Ensemble-DAgger + MET | $0.935 \pm 0.019$ | $2502 \pm 228$ | $0.707 \pm 0.023$ | $77 \pm 4$ | $315 \pm 84$ | $2034 \pm 79$ |
| Ensemble-DAgger | $0.952 \pm 0.018$ | $2489 \pm 108$ | $0.626 \pm 0.045$ | $2785 \pm 41$ | $1452 \pm 134$ | $2871 \pm 94$ |
| RND-DAgger | $0.944 \pm 0.014$ | $2490 \pm 160$ | $0.717 \pm 0.018$ | $368 \pm 11$ | $708 \pm 130$ | $1214 \pm 25$ |
| RND-DAgger w/o MET | - | $2432 \pm 175$ | - | - | $1985 \pm 215$ | - |

Table 4: We conduct an ablation study on the impact of the Minimal Expert Time (MET) on all baselines across our three environments. We report the final task performance and number of context switches. We conducted the same grid search for the value of W as for RND-DAgger (see section A for more details).

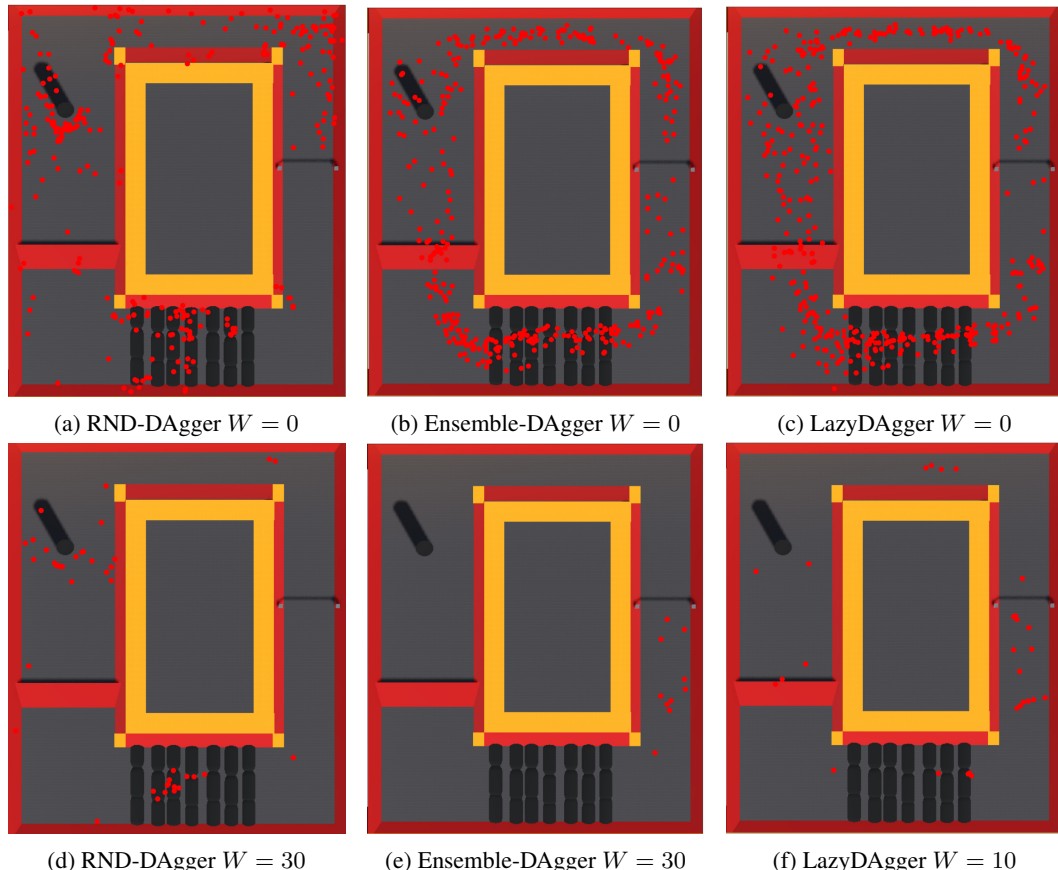

(a) RND-DAgger $W = 0$      (b) Ensemble-DAgger $W = 0$      (c) LazyDAgger $W = 0$

(d) RND-DAgger $W = 30$      (e) Ensemble-DAgger $W = 30$      (f) LazyDAgger $W = 10$

Figure 14: Ablation study: a focus on the effect of the MET on the RaceCar environment. We see an increase of the context switches and shorter episode traces as there is no Minimal Expert Time for all the methods. RND-DAgger seems to have a better grasp on the difficult zones of the track, whereas the two other measures trigger the expert rather uniformly across the track.

## D    TOTAL EXPERT TIME COMPARISONS

While focusing solely on task performance and context switches provides important insights on the efficiency of a method and on a given task, it doesn't tell the complete story about expert burden. We conducted an additional analysis on the **Total Expert Time**, as we did in the Fig 8b on human experiments. The Table 5 shows that Ensemble-Dagger is highly time consuming for the expert without statistically outperforming RND-Dagger. For example, on Maze, it is three times longer to achieve a similar performance than with RND-DAgger, even with the addition of our MET mechanism on that baseline. This demonstrates that we are more efficient in real human expert

scenarios. Moreover, and especially in a fast-paced environment like RaceCar, that baseline - and its improved MET variant - requires the expert to input actions at the same time as the agent to be able to compute the discrepancy measure, which is not natural in practice.

| | Task Performance | | | Total Expert Time | | |
|---|---|---|---|---|---|---|
| | RC | HC | Maze | RC | HC | Maze |
| RND-DAgger | $0.944 \pm 0.014$ | $2490 \pm 160$ | $0.717 \pm 0.018$ | $16664 \pm 73$ ($27.8 \pm 0.1$ mins) | $11968 \pm 107$ ($6.60 \pm 0.06$ mins) | $16173 \pm 44$ ($27.0 \pm 0.1$ mins) |
| Ensemble-DAgger | $0.952 \pm 0.018$ | $2489 \pm 108$ | $0.626 \pm 0.045$ | $31394 \pm 506$ ($52.3 \pm 0.8$ mins) | $21535 \pm 774$ ($11.8 \pm 0.4$ mins) | $51434 \pm 1507$ ($85.7 \pm 2.5$ mins) |
| Ensemble-DAgger + MET | $0.935 \pm 0.019$ | $2502 \pm 228$ | $0.707 \pm 0.023$ | $18216 \pm 165$ ($30.4 \pm 0.3$ mins) | $13474 \pm 631$ ($7.40 \pm 0.35$ mins) | $48125 \pm 900$ ($80.2 \pm 1.5$ mins) |

Table 5: The results considering the **Total Expert Time** i.e. the total number of frames on which the expert had to be involved to provide an action. In the case of Ensemble-DAgger, the expert action is needed at each time step to compute the discrepancy measure between the mean action of the ensemble of policies and the expert action. The time spent in minutes is computed with a typical frame rate of 30 for HalfCheetah (arbitrary) and 60 with one action taken every 6 frames, for the two game environments.

