# OpenReview forum: "Efficient Active Imitation Learning with Random Network Distillation"
_ICLR.cc/2025/Conference — ICLR 2025 Poster_

### Official Review · Reviewer_Qs6v · 2024-10-27

**Soundness:** 2
**Presentation:** 3
**Contribution:** 3
**Rating:** 6
**Confidence:** 3

**Summary:**

The authors address efficient expert intervention in active imitation learning for complex, reward-free tasks like video game bots and robotic control. The proposed RND-DAgger framework uses a state-based out-of-distribution (OOD) measure to trigger expert input selectively, reducing intervention needs by focusing on critical states. Experiments in RaceCar, 3D Maze, and HalfCheetah environments demonstrate that RND-DAgger outperforms traditional methods, achieving high performance with fewer expert queries. Ablation studies highlight the importance of the stability window and historical context in reducing context switches and expert burden.

**Strengths:**

1. **Motivation and Intuition**: The motivation for reducing expert intervention in active imitation learning is convincing, particularly in applications where continuous expert availability is costly or impractical.

2. **Novelty**: The paper introduces the idea of leveraging Random Network Distillation (RND) to manage expert interventions, which is innovative and well-suited for identifying critical OOD states.

3. **Clarity**: The paper is well-organized, with clear explanations of both theoretical foundations and implementation details. Figure 4 effectively illustrates the RND-DAgger algorithm, and ablation studies clarify the effect of each component.

4. **Experimental Results**: The visualizations of experimental results are clear. Figure 6, for instance, provides an accessible overview of task performance and the frequency of context switches across environments, highlighting RND-DAgger’s reduced reliance on expert intervention.

5. **Reproducibility**: The author's inclusion of pseudo-code, a detailed description of the experimental setup, and an appendix for hyperparameters strengthens reproducibility. They promise to release code upon acceptance also supports this.

**Weaknesses:**

1. **Experiment Setup**: The environments in the main experiment, although tested on four different setups, lack sufficient diversity, limiting the evaluation of RND-DAgger’s performance across a broader set of tasks. Incorporating additional environments, such as robot manipulation or hand dexterity tasks, would enrich the setup and test the method’s adaptability in more complex, fine-grained control scenarios.

2. **Related work**: The paper needs to include comparisons to the methods that specifically designed to handle out-of-distribution scenarios, aiming to generalize to states unobserved during training without requiring expert intervention [1, 2].

[1]  Shang-Fu Chen, Hsiang-Chun Wang, Ming-Hao Hsu, Chun-Mao Lai, and Shao-Hua Sun. Diffusion model-augmented behavioral cloning. In International Conference on Machine Learning, 2024.

[2] Pete Florence, Corey Lynch, Andy Zeng, Oscar A Ramirez, Ayzaan Wahid, Laura Downs, Adrian Wong, Johnny Lee, Igor Mordatch, and Jonathan Tompson. Implicit behavioral cloning. In Conference on Robotic Learning, 2022.

**Questions:**

1. The paper lacks necessary comparisons with methods specifically designed to address out-of-distribution scenarios, aiming to generalize to states unobserved during training without requiring expert intervention [1, 2]. Including these comparisons is essential to demonstrate the completeness of the proposed method in handling such challenges and I am willing to raise the scores.

2. While the paper presents results from four environments, the setup may lack sufficient diversity to fully assess RND-DAgger’s adaptability. Would the authors consider evaluating on additional environments, such as robot manipulation or hand dexterity tasks, which could reveal the method’s robustness and effectiveness across a broader range of complex control scenarios?

3. In which types of tasks or environments does RND-DAgger face challenges, especially with regard to state-based out-of-distribution detection?

4. How does the RND mechanism manage computational efficiency, and is it feasible to optimize it further for large-scale applications?

5. What role does the "minimal demonstration time" play in balancing expert workload and agent training quality, especially in complex environments?

[1]  Shang-Fu Chen, Hsiang-Chun Wang, Ming-Hao Hsu, Chun-Mao Lai, and Shao-Hua Sun. Diffusion model-augmented behavioral cloning. In International Conference on Machine Learning, 2024.

[2] Pete Florence, Corey Lynch, Andy Zeng, Oscar A Ramirez, Ayzaan Wahid, Laura Downs, Adrian Wong, Johnny Lee, Igor Mordatch, and Jonathan Tompson. Implicit behavioral cloning. In Conference on Robotic Learning, 2022.

---

> ### Author Response · Authors · 2024-11-22
> **Answer to R-Qs6v**
>
> We thank R-Qs6v for their thorough review. We appreciated learning that the motivation of our work was found convincing and that our proposed RND-DAgger approach was judged innovative.
>
> > R-Qs6v:”While the paper presents results from four environments, the setup may lack sufficient diversity to fully assess RND-DAgger’s adaptability. Would the authors consider evaluating on additional environments, such as robot manipulation or hand dexterity tasks, which could reveal the method’s robustness and effectiveness across a broader range of complex control scenarios?”
>
> While we agree with the suggestion that evaluating RND-DAgger in a broader range of environments could enrich our experimental analysis, we believe that the current selection of environments – spanning a robotic Half-Cheetah domain, a goal-conditioned navigation maze and a car racing game – already offers sufficient diversity. These setups encompass tasks with varying dynamics, observations, objectives, action spaces and levels of complexity, providing a well-rounded evaluation of RND-DAgger’s performance.
>
> > R-Qs6v:”The paper lacks necessary comparisons with methods specifically designed to address out-of-distribution scenarios, aiming to generalize to states unobserved during training without requiring expert intervention [1, 2]. Including these comparisons is essential to demonstrate the completeness of the proposed method in handling such challenges and I am willing to raise the scores.”
>
> We thank R-Qs6v for mentioning these two interesting works, that will be added in the updated manuscript in the related work section. We kindly refer R-Qs6v to our “answer to all reviewers” post, section “Considered Baselines” for details and positioning over existing related work that were missing in the initial submission.
>
> As mentioned in our “answer to all reviewer” post, these two works ([1,2] of R-Qs6v)  present improved Imitation Learning approaches compared to standard approaches such as Behavioral Cloning. These works do not address the interactive imitation learning problem, i.e. how to better leverage expert interventions to accelerate data collection and improve performances. We have demonstrated in our paper that **it is possible to include OOD measures for interactive imitation learning** which is as far as we know the first result in this direction. Testing for different OOD measures is a good research direction that we expect to tackle as a future work thanks to the R-Qs6v's remarks.
>
>
> > R-Qs6v:”In which types of tasks or environments does RND-DAgger face challenges, especially with regard to state-based out-of-distribution detection?
>
> The current approach would most likely struggle in environments featuring noisy-TV problems, i.e. environments featuring distracting random observations. In such cases, such random features could be filtered through representation learning (or explicit high-entropy filtering), or the RND-based OOD detection could be combined with a forward model prediction loss. While we believe this topic to be out of scope for the present work, it underlines an interesting line of future work. We thank R-Qs6v for pointing out the lack of discussion on RND-DAgger limitations. We will update the conclusion section of the manuscript to include this aforementioned noisy-TV limitation.
>
>
> > R-Qs6v:”How does the RND mechanism manage computational efficiency, and is it feasible to optimize it further for large-scale applications?
>
> RND-DAgger does provide a computationally efficient alternative to approaches such as Ensemble-DAgger: instead of an ensemble of policies that must be trained and used at inference time, we only need to train a single policy and a predictor, and inference time requires these two components, along with the fixed random network. For large-scale applications, RND-DAgger is a particularly well-suited interactive approach compared to DAgger and HG-DAgger which require constant expert supervision: with RND-DAgger multiple agents can be evaluated in parallel (in accelerated simulations), with the expert only allocating time when one agent triggers expert intervention.
>
> > R-Qs6v:”What role does the "minimal demonstration time" play in balancing expert workload and agent training quality, especially in complex environments?
>
>
> Given similar concerns were raised by other reviewers (R-r6sZ,R-dV7L), we kindly refer R-Qs6v  to our “answer to all reviewers” post, section “On the impact of Minimal Expert Time”. By Minimal Expert Time we refer to this minimal demonstration time (we will be more consistent with naming in the updated manuscript).

---

> > ### Comment · Reviewer_Qs6v · 2024-11-24
> >
> > Thank you for your thorough response to my questions. It effectively resolved my main concerns, so I’ve raised my rating to 6. Your explanation regarding [1,2] is reasonable, and I look forward to the updated manuscript.

---

> ### Author Response · Authors · 2024-12-01
> **Answer to R-Qs6v (follow-up)**
>
> We sincerely thank Reviewer Qs6v for the constructive discussion, which has significantly contributed to improving our manuscript. As the discussion period concludes, we would like to take this opportunity to follow up on your feedback.
>
> Regarding experimental significance, as previously mentioned we conducted additional experiments described in our “answer to all reviewers” post. This post now includes an analysis of EnsembleDAgger vs RND-DAgger regarding how much **total expert time** is needed during interactive training (section “answer about experiments”). Overall, these experiments confirm RND-DAgger as a promising interactive learning method, which **outperforms EnsembleDAgger and LazyDAgger regarding expert burden and/or task performance** in all considered environments.
>
> Our updated manuscript now features this new or modified content (red text, e.g. section 5, appendix C and D). We are also committed to incorporating the discussion of related work presented in our “Answer to all reviewers” post, section “considered baselines” as an additional appendix in the final manuscript.
>
> We hope these updates contribute to a more comprehensive evaluation of our work and would greatly appreciate it if you could reconsider your score in light of these improvements. We remain open to further discussion and suggestions to further improve the paper.
>
> Thank you once again for your time and effort throughout this process.

---

### Official Review · Reviewer_dV7L · 2024-10-31

**Soundness:** 3
**Presentation:** 3
**Contribution:** 2
**Rating:** 6
**Confidence:** 3

**Summary:**

The paper focuses on addressing the covariate-shift problem in behavioural cloning, where the agent suffers from compounding error from the predicted actions. The paper considers DAgger-like algorithms where the process includes querying an expert to add extra demonstrations, thereby increasing state-action coverage. The paper identifies a gap where existing approaches require the said expert to be present for a long period of time, which can be costly for some applications. The paper thus propose a method that considers out-of-distribution in states through random network distillation. The paper conducts experiments on three simulated environments, demonstrating that the proposed algorithm achieves similar performance as existing DAgger-like baselines, as well as the reduced number of expert queries.

**Strengths:**

- The idea is very simple and focuses on a gap that previous approaches do not consider (i.e. OOD states rather than action mismatch).
- The algorithm appears to be practical compared to existing approaches.

**Weaknesses:**

I am happy to increase the scores if these comments are addressed.

**Comments**
- I am unsure why the formulation is POMDP rather than MDP. The environments used in this paper appears to be using state-based information, in the sense that I cannot really guarantee that they are partially-observable, as opposed to using images that will be way more convincing.
	- I further believe that the current method assumes the observation aliasing is not a problem. Self-driving environment can dramatically different scene while similar actions.
	- I suggest including the CARLA environment to make this result more convincing.
- Regarding the approach, experimentally how is this algorithm different from the earlier variants since $f_{targ}$ can be $\pi_{exp}$? In other words, is LazyDAgger essentially doing RND but have a slightly higher-dimensional output space? If that is the case, then is the benefit coming from the additional "minimal demonstration time" mechanism?
- The paper can improve upon its writing quality:
	- Algorithms 3 and 4: Include the actual definition of *measure* for clarity.
	- Table 1: What do bolded and underlined terms mean? Is this taking the average or some other statistics? What about the standard error or other statistics?
	- Figure 6: What do the solid line and shaded area correspond to?
		- Also recommend the top curve to use different line style to differentiate variants easier.

**Questions:**

- Experiments:
	- In Figure 6, do the algorithms use same initial datasets? Does increasing in x-axis mean starting at a larger dataset? Or this corresponds to increasing size due to the conditions to query from the expert?
	- How do Figure 6e and 6f demonstrate that RND-DAgger is better in "sample-efficiency"? Context switching does not totally correspond to number of samples queries from the expert. Can the paper clarify this point?
	- On page 9, there is a claim "..., RND-DAgger is more effective at leveraging expert feedback to improve the policy." How is this statement true exactly? The policy training at the end is the same but I suppose the aggregated data is somewhat different?

---

> ### Author Response · Authors · 2024-11-22
> **Answer to R-dV7L (1/2)**
>
> We thank R-dV7L for their review. We were pleased to learn that our work was seen as filling a gap in the interactive imitation learning literature and that our proposed algorithm appeared practical compared to existing methods.
>
> > R-dV7L:” I am unsure why the formulation is POMDP rather than MDP. The environments used in this paper appears to be using state-based information, in the sense that I cannot really guarantee that they are partially-observable, as opposed to using images that will be way more convincing.”
>
> It is true that the use of a POMDP formulation encompasses a wider range of scenarios than those considered in our experimental section, which are two state-based MDPs and goal-conditioned state-based MDPs. We believe current experiments, which encompass a wide range of state-based scenarios (racing, goal-conditioned maze navigation, and robotic locomotion) are sufficient to demonstrate the advantages of RND-DAgger. We note that our racing and maze environments include depth map images, built through ray-casts. As pointed out by R-USwa, we believe our current environment set is sufficient to demonstrate the advantages of our approach. We leave the (definitely interesting) study of image-based environments to future work.
>
> > R-dV7L:” I further believe that the current method assumes the observation aliasing is not a problem. Self-driving environment can dramatically different scene while similar actions.”
>
> The current approach would indeed most likely struggle in environments featuring noisy-TV problems, i.e. environments featuring distracting random/diverse observations. In such cases, such random features could be filtered through representation learning (or explicit high-entropy filtering), or the RND approach could be combined with a forward model prediction loss. While we believe this topic to be out of scope for the present work, it underlines an interesting line of future work. We thank R-dV7L for pointing out the lack of discussion on RND-DAgger limitations. We will update the conclusion section of the manuscript to include this aforementioned noisy-TV/observation aliasing limitation.
>
> > R-dV7L: I suggest including the CARLA environment to make this result more convincing.
>
> As aforementioned in our two previous answers, we believe our current environment set is sufficient to demonstrate our method. While we thank R-dV7L for pointing out the CARLA self-driving environment, we believe it would be better suited to a follow-up future work focused on the study of RND-DAgger scaling to more complex pixel-based settings with observation aliasing challenges.
>
> > R-dV7L:”Regarding the approach, experimentally how is this algorithm different from the earlier variants since f_targ can be pi_exp ? In other words, is LazyDAgger essentially doing RND but have a slightly higher-dimensional output space? If that is the case, then is the benefit coming from the additional "minimal demonstration time" mechanism?”
>
> LazyDAgger does rely on an error prediction to trigger intervention like RND-DAgger. However, this error prediction is done over the action space. While this does change the dimensionality of the vector to be predicted, we argue the most important difference is that predicting actions is less reliable with human experts, which can have stochastic behaviors (many actions can lead to the same overall state sequence). Additionally, LazyDAgger relies on defining a hysteresis band, which we do not use, while we use our Minimal Expert Time mechanism.
>
> Clarifications on the impact of the minimal demonstration time (Minimal Expert Time) hyperparameter have been raised by multiple reviewers. As such, we kindly refer R-dV7L to our “answer to all reviewers” post where the “On the impact of Minimal Expert Time” section answers these concerns and provides new experiments to further support our claims.

---

> ### Author Response · Authors · 2024-11-22
> **Answer to R-dV7L (2/2)**
>
> > R-dV7L (writing quality):”Algorithms 3 and 4: Include the actual definition of measure for clarity.”
>
> Thanks for pointing this out, these measures are defined in the caption of figure 3, but it is true that adding them in the algorithm would make it clearer. We will do so in the updated manuscript.
>
> > R-dV7L (writing quality):”Table 1: What do bolded and underlined terms mean? Is this taking the average or some other statistics? What about the standard error or other statistics?
>
> In Table 1 we show the average post-training performance over 8 seeds (task performance) along with the average expert cost (# of context switching) over the same 8 seeds. The top values per environment and metric are bolded, and the second one is underlined. We will add this information to the caption of Table 1.
> Following your question about other statistics, which is similar to concerns raised by R-USwa, we added 95% confidence intervals to table 1:
> | Method       	| RC Task Performance  | HC Task Performance   | Maze Task Performance  | RC Context Switch | HC Context Switch  | Maze Context Switch |
> |-------------------|-------------------------------|-------------------------------|-------------------------------|---------------------------|---------------------------|-----------------------------|
> | **BC**       	| 0.883 $\pm$ 0.029        	| 2455 $\pm$ 424           	| 0.367 $\pm$ 0.043        	| -                     	| -                     	| -                       	|
> | **DAgger**   	| 0.940 $\pm$ 0.026        	| 2343 $\pm$ 209           	| 0.450 $\pm$ 0.074        	| -                     	| -                     	| -                       	|
> | **Lazy-DAgger**  | 0.939 $\pm$ 0.025        	| 2314 $\pm$ 278           	| 0.575 $\pm$ 0.101        	| 3437 $\pm$ 89        	| **312 $\pm$ 45**      	| _1238 $\pm$ 79_        	|
> | **Ensemble-DAgger** | **0.952 $\pm$ 0.018**  	| _2489 $\pm$ 108_         	| _0.626 $\pm$ 0.045_      	| _2785 $\pm$ 41_      	| 1452 $\pm$ 134        	| 2871 $\pm$ 94          	|
> | **RND-DAgger**   | _0.944 $\pm$ 0.014_      	| **2490 $\pm$ 160**       	| **0.717 $\pm$ 0.018**    	| **368 $\pm$ 11**     	| _708 $\pm$ 130_       	| **1214 $\pm$ 25**      	|
>
>
> > R-dV7L (writing quality):”Figure 6: What do the solid line and shaded area correspond to?
>
> The solid line represents the mean performance (or context switch) over 8 seeds, the shaded area is the standard error of the mean.
>
> > R-dV7L:”How do Figure 6e and 6f demonstrate that RND-DAgger is better in "sample-efficiency"? Context switching does not totally correspond to number of samples queries from the expert. Can the paper clarify this point?”
>
> We agree that figures 6e and 6f do not enable this sample efficiency claim.
> The sample-efficient nature of RND-DAgger training is better assessed using Figure 6 top row (6a,6b,6c), i.e. observing how RND-DAgger outperforms baselines such as LazyDAgger along training (especially on 6b and 6c). For similar dataset sizes (the x-axis), RND-DAgger outperforms LazyDAgger: it is sample efficient. We will update the experimental section to clearly mention subfigures to support our claims (rather than just “figure 6”).
>
>
> > R-dV7L:” On page 9, there is a claim "..., RND-DAgger is more effective at leveraging expert feedback to improve the policy." How is this statement true exactly? The policy training at the end is the same but I suppose the aggregated data is somewhat different?”
>
> We acknowledge that our phrasing could be improved here. Indeed, aggregated data throughout expert interventions is different for each approach (and each seed). What we meant here (in the context of the Maze task) is that, given that LazyDagger has a comparable #context switch to RND-Dagger (fig 6f) but that RND-DAgger outperforms Lazy-Dagger (fig 6e), RND-Dagger intervention criterion lead to more informative expert interventions. In other words, it is better able to leverage the expert. We will clarify this in the updated manuscript (as for the previous comment, mentioning subfigures alongside claims will help).

---

> > ### Comment · Reviewer_dV7L · 2024-11-24
> >
> > Hi, thank you for the response. The authors have addressed few questions but the critical ones aren't totally addressed. As a result I have kept my score.
> >
> > 1. I am still concerned about the inclusion of using POMDP formulation when the experiments aren't reflecting the same setting---again I believe having some image-based (if not truly POMDP envrionments like image key-door environment in grid world) results will be more convincing. I strongly suggest removing more general formulation and be very open about the limitations when the algorithm is currently not validated to perform well in these situations, which can be very misleading for researchers to reproduce the proposed method.
> >
> > 2. I agree with reviewer r6sZ that the additional results are not demonstrating statistically significant advantage over baseline approaches.
> >
> > 3. Regarding stochasticity of the expert, that may be true---I suggest the paper describe this point further.

---

> > > ### Author Response · Authors · 2024-11-26
> > > **Answer to R-dV7L**
> > >
> > > We thank R-dv7L for their reply.
> > >
> > > About the POMDP formulation: following your suggestion, we will define our approach within an MDP setting for clarity. We now mention the importance of studying the approach in POMDP and pixel-based settings in our updated “Conclusion and limitations” section 5 of the updated manuscript.
> > >
> > > Regarding our experiments, we kindly refer R-dv7L to the new section of our “answer to all reviewer" post, named “answer about experiments”

---

> ### Comment · Reviewer_dV7L · 2024-11-26
>
> I appreciate the authors' response. They have addressed my concerns and I have increased the score. My final request is to include the expert-time metric in the main paper or emphasize this in the main paper with a reference to the appendix, as well as making sure the POMDP changes are completed.

---

> > ### Author Response · Authors · 2024-11-27
> > **Answer**
> >
> > We are pleased to see that our answers solved your concerns.
> >
> > > My final request is to include the expert-time metric in the main paper or emphasize this in the main paper with a reference to the appendix, as well as making sure the POMDP changes are completed.
> >
> > Following your suggestion, we updated the manuscript to follow an MDP formulation, i.e. updating our preliminaries: see "Notations" paragraph in section 2 of the updated manuscript.
> >
> > We now mention expert-time metric experiments (Appendix D) clearly in an “additional experiments” paragraph, added at the end of the updated section 4.
> >
> > We would like to thank R-dV7L for the fruitful discussion, which helped us improve the present work.

---

> ### Author Response · Authors · 2024-12-01
> **Answer to R-dV7L (follow-up)**
>
> We thank R-dV7L for their detailed feedback and engagement throughout the review process. Your constructive suggestions have greatly contributed to improving our manuscript.
>
> We hope our previously mentioned updates along with our new experiments described in our “answer to all reviewers” addressed your remaining concerns and would greatly appreciate it if you could reconsider your score in light of these improvements. If you have any additional feedback or suggestions, we would be delighted to address them promptly.
>
> Thank you for your time

---

### Official Review · Reviewer_r6sZ · 2024-11-02

**Soundness:** 3
**Presentation:** 2
**Contribution:** 2
**Rating:** 6
**Confidence:** 4

**Summary:**

This paper presents Random Network Distillation DAgger (RND-DAgger), an active imitation learning method that leverages RND to define an out-of-distribution states, enabling selective expert intervention and minimize the frequency of transitions between human experts and learning agent through a minimal demonstration time mechanism. The approach is evaluated across Race Car, Maze, and Half Cheetah environments.

**Strengths:**

- The paper presents a well-founded motivation, aiming to optimize the timing of expert interventions to reduce overall costs associated with human expertise and minimize the frequency of transitions between human experts and learning agent.
- The paper is straightforward, particularly for readers with a background in the domain of unsupervised RL.

**Weaknesses:**

- **Novelty:** The paper's novelty appears constrained, as it predominantly builds upon an established novelty measure within the unsupervised RL domain. This integration approach mirrors Ensemble-Dagger, which relies on a principle similar to Disagreement in the unsupervised RL domain.

- **Limitations:** The limitations specific to this method are not sufficiently addressed. Certain limitations noted by the authors for other approaches may also apply here. For instance, the paper states, “While this approach works well when the expert is optimal and acts deterministically, it becomes problematic when dealing with humans or imperfect experts.” It is unclear how this method manages these challenges, and if not addressed, this issue should be explicitly acknowledged.

- **Baselines:** The study lacks comparisons with more recent methods that similarly leverage human input for out-of-distribution states, such as RLIF [1], PATO [2], and Sirius [3].

- **Time Mechanism:** This approach is reliant on two factors: RND and Minimal Expert Time. It would be valuable to assess how Minimal Expert Time operates alongside other baseline methods - a potentially straightforward inclusion. This would provide a clearer understanding of RND’s advantages over alternative metrics.

- **Experiments:** Additional experiments are needed to support the authors' claims regarding task performance and context switching. From the current results (Table 1), Ensemble-Dagger performs better in RC settings but with higher context switches, whereas Lazy Dagger shows the opposite trend in HC settings. Further experiments in more challenging environments, such as Adroit [4], would provide deeper insights and strenghten the paper’s claims. Moreover, incorporating the Time Mechanism into other methods may enhance their context-switching capabilities, offering a compelling comparison.

- **Ablation Study:** The ablation study section would benefit from greater detail, as the current presentation makes it challenging to interpret the impact of various factors on learning outcomes.

**References**

[1] Luo, Jianlan, et al. "RLIF: Interactive Imitation Learning as Reinforcement Learning."arXiv preprint arXiv:2311.12996 (2023).

[2] Dass, Shivin, et al. "Pato: Policy assisted teleoperation for scalable robot data collection."arXiv preprint arXiv:2212.04708 (2022).

[3] Liu, Huihan, et al. "Model-based runtime monitoring with interactive imitation learning."2024 IEEE International Conference on Robotics and Automation (ICRA). IEEE, 2024.

[4] Justin Fu, Aviral Kumar, Ofir Nachum, George Tucker, and Sergey Levine. D4rl: Datasets for deep data-driven reinforcement learning. arXiv preprint arXiv:2004.07219.

**Questions:**

They are mentioned in Weaknesses.

---

> ### Author Response · Authors · 2024-11-22
> **Answer to R-r6sZ**
>
> We thank R-r6sZ for their review. We appreciated that the motivation of our work (minimizing expert intervention in interactive imitation learning) was seen as well-founded and that R-r6sZ found our paper straightforward to read.
>
>
> > R-r6sZ: “The paper's novelty appears constrained”
>
> Given this comment was raised by multiple reviewers, we kindly refer R-r6sZ to our “answer to all reviewers” post where we propose a detailed analysis of why we believe our work meets novelty criteria.
>
> > R-r6sZ: “Certain limitations noted by the authors for other approaches may also apply here. For instance, the paper states, “While this approach works well when the expert is optimal and acts deterministically, it becomes problematic when dealing with humans or imperfect experts.” It is unclear how this method manages these challenges, and if not addressed, this issue should be explicitly acknowledged.”
>
> We apologize for the lack of clarity. What we meant with this claim is that our approach uses state discrepancy to trigger expert interventions, which is more robust than action discrepancy in complex environments where multiple actions can lead to the same state. Action-based methods can trigger unnecessary interventions when different actions achieve similar outcomes, whereas state-based methods focus on meaningful divergences. We will update this sentence in the manuscript to clarify this point.
>
> > R-r6sZ: The study lacks comparisons with more recent methods that similarly leverage human input for out-of-distribution states, such as RLIF [1], PATO [2], and Sirius [3].
>
> Given other reviewers raised concerns about the baselines we consider, we kindly refer R-r6sZ to our “answer to all reviewers”, section “Considered Baselines”, where we position our work w.r.t. suggested baselines.
>
> > R-r6sZ:“This approach is reliant on two factors: RND and Minimal Expert Time. It would be valuable to assess how Minimal Expert Time operates alongside other baseline methods - a potentially straightforward inclusion. This would provide a clearer understanding of RND’s advantages over alternative metrics.”
>
> We thank the reviewer for this suggestion, which we followed. We kindly refer R-r6sZ to our “answer to all reviewers” post where these additional experiments are presented and discussed in the “On the impact of Minimal Expert Time” section.
>
>
> > R-r6sZ:“Additional experiments are needed to support the authors' claims regarding task performance and context switching. From the current results (Table 1), Ensemble-Dagger performs better in RC settings but with higher context switches, whereas Lazy Dagger shows the opposite trend in HC settings.
>
> Regarding Table 1 results, following R-USwa suggestion to add confidence intervals, we can now observe that Ensemble-DAgger performance on Race Car is comparable to RND-DAgger, there is no statistically significant difference (p=0.23 using Welch’s student t-test).
>
> For Half-Cheetah, while the post-training number of context is lower for Lazy-DAgger, figure 6b showcases that **RND-DAgger outperforms Lazy-DAgger regarding task performance along training**, i.e. it is more expert-data sample efficient. For instance, it is statistically significant with p=0.03 for a dataset size of 4000 (x-axis value in fig 6b) using a Welch’s student t-test. In Half-Cheetah, given that all baselines saturate at similar post-training task performances (fig 6b), sample efficiency is a representative measure.
>
>
> > R-r6sZ:”Further experiments in more challenging environments, such as Adroit [4], would provide deeper insights and strenghten the paper’s claims.
>
> While we agree with the suggestion that evaluating RND-DAgger in a broader range of environments could enrich our experimental analysis, we believe that the current selection of environments – spanning a robotic Half-Cheetah domain, a goal-conditioned navigation maze and a car racing game – already offers sufficient diversity. These setups encompass tasks with varying dynamics, observations, objectives, action spaces and levels of complexity, providing a well-rounded evaluation of RND-DAgger’s performance.
>
> > R-r6sZ:”incorporating the Time Mechanism into other methods may enhance their context-switching capabilities, offering a compelling comparison.”
>
> We thank R-r6sZ for proposing these **additional experiments, which we conducted**. We kindly refer R-r6sZ to our “answer to all reviewer” post, section “On the impact of Minimal Expert Time”

---

> > ### Comment · Reviewer_r6sZ · 2024-11-23
> > **Comments on the Rebuttal**
> >
> > I appreciate the effort from the authors in addressing my concerns. Thank you for your thorough responses, additional experiments and related modifications to the paper.
> >
> > Regarding the MET criterion, I believe it is crucial to note that Ensemble-Dagger performs almost similarly in two environments and slightly outperforms RND-DAgger in HC, while the context switches are fewer in the two environments. However, the authors' point about the reliance on expert action remains valid. Empirically, the results appear mixed, suggesting that testing on more environments is necessary to draw a robust conclusion about the performance of RND-DAgger.
> >
> > I also thank the authors for their clarification on the baseline methods. Those points are valid and convincing.
> >
> > At this stage, I would prefer to keep my score unchanged. While the idea is promising in general, it is currently difficult to observe any significant advantage over the baselines. Additional results across diverse environments would certainly strengthen the case.

---

> > > ### Author Response · Authors · 2024-11-26
> > > **Answer to R-r6sZ**
> > >
> > > We thank R-r6sZ for their reply. Regarding the significance of our experiments, we kindly refer R-r6sZ to the new section of our “answer to all reviewer” post, last section “answer about experiments”.

---

> ### Author Response · Authors · 2024-12-01
> **Answer to R-r6sZ (follow-up)**
>
> We thank Reviewer r6sZ for the constructive discussion. As the discussion period comes to a close, we would like to take this opportunity to follow up.
>
> Regarding experimental significance, we understand that R-r6sZ's main concern is related to Ensemble-DAgger vs RND-DAgger. As previously mentioned, we conducted additional experiments described in our “answer to all reviewers” post section “Answer about experiments”, which features an analysis of EnsembleDAgger vs RND-DAgger regarding how much **total expert time** is needed during interactive training. This additional perspective better showcases that **Ensemble-DAgger is highly time consuming for the expert without statistically outperforming RND-Dagger** (even when augmenting it with our MET mechanism).
>
> We hope these updates contribute to a more comprehensive evaluation of our work and would greatly appreciate it if you could reconsider your score in light of these improvements. We remain open to further discussion and suggestions to further improve the paper.
>
> Thank you again for your time.

---

> ### Comment · Reviewer_r6sZ · 2024-12-02
> **Comments on the Rebuttal (2)**
>
> I thank the authors for their timely response and additional details provided. I have also read other reviewer's comments and authors responses. I am changing my score to "Above Threshold".

---

### Official Review · Reviewer_USwa · 2024-11-04

**Soundness:** 3
**Presentation:** 3
**Contribution:** 2
**Rating:** 6
**Confidence:** 4

**Summary:**

This work presents a method to more effectively integrate human feedback with imitation learning and specifically the IL method dagger. The idea is to only request human feedback when the policy is out of distribution. The data collected by the expert is then added to a buffer for training. In doing so, the algorithm is able to more effectively utilize the experts time and feedback.

**Strengths:**

The research problem this work focuses on is good. Increasing the efficiency of imitation learning is an open problem and the most effective way to utilize expert data within IL. Improving the interaction between expert and policy increases the ability of IL to be used with real world problems.

The contribution of this work is ok, they propose their method and perform a study to verify their claims.

The algorithm is clearly defined and could be implemented from the information given.

The experiments run are reasonable and seem to demonstrate their method well.

The baseline comparison are all dagger variants (and BC). They are definitely reasonable comparisons.

The empirical results are good. They show a balance between expert interventions and performance which is the goal of this work.

The clarity is great. This paper was very easy to read and very clear.

**Weaknesses:**

The novelty of this work is minimal. As far as I understand it the methods generally used in this work are all previously known. Dagger and the OOD classifier seem to be like the main components used but are previous work.

The statistical rigor needs improvement. The metrics are only averaged over 8 seeds. Is there a reason for only this many? I feel like it should be many more.

As well, I want to see confidence intervals on Table 1.

There is no failure analysis. I wish there was one on the times the method does not perform better than the previous methods. What about those tasks makes it not as good?

The future work and conclusion is ok. I wish there was a better future work section. I think a limitation of this work is that it works well in simulated environments but on an actual self-driving car how would this work? The car couldn’t just stop running. This could be an interesting direction of future work some like predicting beforehand that you’re going to get an out of distribution state and request input.

**Questions:**

I wonder are there other comparisons here? Do they all have to be dagger variants? Is there another type of “expert takeover” method that you could compare to? If I missed this in the text I apologize.

---

> ### Author Response · Authors · 2024-11-22
> **Answer to R-USwa (1/2)**
>
> We thank R-USwa for their detailed review. We were pleased to see that R-USwa identified 1) the problem of efficient imitation learning as a good research direction, 2) found our manuscript to be clearly written, and 3) deemed our contribution and its experimental analysis to be reasonable.
>
> > R-USwa:“The novelty of this work is minimal.”
>
> Given this comment was raised by multiple reviewers, we kindly refer R-USwa to our “answer to all reviewers” post, first section, where we detail why we believe our work meets novelty criteria.
>
> > R-USwa:“The statistical rigor needs improvement. The metrics are only averaged over 8 seeds. Is there a reason for only this many? I feel like it should be many more. As well, I want to see confidence intervals on Table 1.”
>
> While we agree that featuring more seeds is always beneficial, we believe using 8 seeds is a typical choice. Multiple papers already published in top-tier conferences have been using 8 seeds or less, e.g.
>
> Park, S., Ghosh, D., Eysenbach, B., & Levine, S. Hiql: Offline goal-conditioned rl with latent states as actions (Neurips 2024). → 8 seeds
>
> Chen, L., Lu, K., Rajeswaran, A., Lee, K., Grover, A., Laskin, M., ... & Mordatch, I. Decision transformer: Reinforcement learning via sequence modeling. (Neurips 2021) → 3 seeds
>
>  In the LazyDAgger paper, one of our baselines, the original authors use 3 seeds. That being said, we agree that our experimental analysis could be complemented with additional statistical analysis to strengthen our claims. Following your comment, in the updated manuscript we updated Table 1 with 95% confidence intervals, as follows
>
> | Method       	| RC Task Performance  | HC Task Performance   | Maze Task Performance  | RC Context Switch | HC Context Switch  | Maze Context Switch |
> |-------------------|-------------------------------|-------------------------------|-------------------------------|---------------------------|---------------------------|-----------------------------|
> | **BC**       	| 0.883 $\pm$ 0.029        	| 2455 $\pm$ 424           	| 0.367 $\pm$ 0.043        	| -                     	| -                     	| -                       	|
> | **DAgger**   	| 0.940 $\pm$ 0.026        	| 2343 $\pm$ 209           	| 0.450 $\pm$ 0.074        	| -                     	| -                     	| -                       	|
> | **Lazy-DAgger**  | 0.939 $\pm$ 0.025        	| 2314 $\pm$ 278           	| 0.575 $\pm$ 0.101        	| 3437 $\pm$ 89        	| **312 $\pm$ 45**      	| _1238 $\pm$ 79_        	|
> | **Ensemble-DAgger** | **0.952 $\pm$ 0.018**  	| _2489 $\pm$ 108_         	| _0.626 $\pm$ 0.045_      	| _2785 $\pm$ 41_      	| 1452 $\pm$ 134        	| 2871 $\pm$ 94          	|
> | **RND-DAgger**   | _0.944 $\pm$ 0.014_      	| **2490 $\pm$ 160**       	| **0.717 $\pm$ 0.018**    	| **368 $\pm$ 11**     	| _708 $\pm$ 130_       	| **1214 $\pm$ 25**      	|
>
>
>
> > R-USwa:“there is no failure analysis. I wish there was one on the times the method does not perform better than the previous methods. What about those tasks makes it not as good?”
>
> Thanks to R-USwa suggestions, the addition of confidence intervals allows us to better understand task performance regarding our experiments: one can see in Table 1 that there is no prior method that is statistically significantly superior to RND-DAgger regarding task performance. RND-DAgger is even statistically significantly superior to previous methods on the Maze task performance. Most importantly, given equal performances  RND-DAgger is statistically significantly superior to other methods in the Race Car environment regarding the number of context switches. For Half-Cheetah, while the post-training number of context is lower for Lazy-DAgger, figure 6b showcases that RND-DAgger outperforms Lazy-DAgger regarding task performance along training, i.e. it is more expert-data sample efficient. For instance, it is statistically significant with p=0.03 for a dataset size of 4000 (x-axis value in fig 6b) using a Welch’s student t-test. In Half-Cheetah, given that all baselines saturate at similar post-training task performances (fig 6b), sample efficiency is a representative measure.

---

> ### Author Response · Authors · 2024-11-22
> **Answer to R-USwa (2/2)**
>
> > R-USwa: “I think a limitation of this work is that it works well in simulated environments but on an actual self-driving car how would this work? The car couldn’t just stop running. This could be an interesting direction of future work some like predicting beforehand that you’re going to get an out of distribution state and request input.”
>
>
> We thank R-USwa for pointing out this missing limitation discussion. We agree that in this work we assume it is feasible to let the learning agent self-trigger expert demonstration, which enables expert-data-efficient training and enables to speed up the data collection process (e.g. accelerating simulator time or parallelizing setup). The current limitation of this approach is that in some real-world scenarios, it is not feasible to stop or pause “until the expert is ready to take over”, e.g. autonomous driving or complex robotics manipulation. As suggested by R-USwa, in such scenarios working on predictive systems could be an interesting solution.
>
>
> > R-USwa:“I wonder are there other comparisons here? Do they all have to be dagger variants? Is there another type of “expert takeover” method that you could compare to? If I missed this in the text I apologize”
>
> This question relates to other novelty concerns raised by R-USwa and other reviewers. We kindly refer to R-USwa to our main answer, which details why other baselines were not considered.

---

> > ### Comment · Reviewer_USwa · 2024-11-23
> >
> > Thank you for your response.
> >
> > In response to the novelty. I believe that your response is reasonable in combining previous methods is a valid contribution as long as the results are impactful to the field. I'm going to leave the contribution as fair because I would disagree with your analysis of the data with the updated confidence intervals. It seems that your method does not outperform the other methods in a statistically significant fashion except for on the maze task. It then has less context switches than lazy-dagger in one task but that result is flipped in the other and they are the same in the last. Seeing this, I think that you need to add more analysis to the paper and possibly other experiments showing the specific type of setting (experiments similar to the maze) where your method outperforms others. I think it is valid to say "our method outperforms in the domain of _____" but I would like to see empirical proof of that and a complete analysis on how that has impacts on the literature.
> >
> > You may also be able to tighten confidence intervals by running more experiments and therefore outperform other methods in a statistically significant fashion but that is up to you if it is not feasible to do this.
> >
> > I would also still like the limitations to be addressed in the full paper. They don't have to be my suggestions but the authors should add that.

---

> > > ### Author Response · Authors · 2024-11-26
> > > **Answer to Reviewer USwa**
> > >
> > > We thank R-USwa for their reply.
> > >
> > > We are happy to share an updated conclusion and limitations section 5 in the updated manuscript, where previously discussed limitations are now mentioned.
> > >
> > > Given the significance of our experiments was challenged by other reviewers, we kindly refer R-USwa to the new section of our “answer to all reviewer" post, named “answer about experiments”.

---

> ### Author Response · Authors · 2024-12-01
> **Answer to R-USwa (follow-up)**
>
> We thank Reviewer USwa for the constructive discussion. As the discussion period comes to a close, we would like to take this opportunity to follow up.
>
> As requested, we did update our conclusion (now “conclusion and limitations”) to discuss important limitations. Regarding experimental significance, we conducted additional experiments described in our “answer to all reviewers” post, including an analysis of EnsembleDAgger vs RND-DAgger regarding how much total expert time is needed during interactive training (section answer about experiments). Overall, these experiments confirm RND-DAgger as a promising interactive learning method, which **outperforms EnsembleDAgger and LazyDAgger regarding expert burden and/or task performance** in all considered environments. In Half-Cheetah a method’s performance is better assessed throughout training as all methods tend to converge by the end of training (see fig 6b) and 13b)).
>
> We hope these updates contribute to a more comprehensive evaluation of our work and would greatly appreciate it if you could reconsider your score in light of these improvements. We remain open to further discussion and suggestions to further improve the paper.
>
> Thank you again for your time.

---

### Author Response · Authors · 2024-11-22
**Answer to all reviewers (1/3)**

We thank all reviewers for their valuable comments on the paper. We have addressed each reviewer individually, here we highlight key points and changes in the paper.

# Novelty


Reviewers R-USwa and R-r6sZ question the novelty of the approach. The model we propose is built on top of recent advances and describes a new active imitation learning (a.k.a. interactive imitation learning) method based on three components:

* The adaptation of the DAgger algorithm to handle  out-of-distribution (OOD) criteria,

* The use of Random Network Distillation as a criterion to detect OOD samples,

* The use of a Minimal Expert Time (MET) to smooth the interventions of the experts.

We agree that none of these individual ingredients are novel, but **the novelty comes from the combination of the ingredients** in a totally different context: active imitation learning. For instance, Random Network Distillation has been developed for favoring exploration in online reinforcement learning but **has never been investigated in the problem of active imitation learning** which we consider to be a real novelty. We note that **R-Qs6v** found our proposed approach innovative, and **R-dV7L** mentioned it as **filling a gap** in the literature. In addition, motivated by concrete industrial problems (e.g. the development of bots in video games), our article proposes new environments specific to video games. As such, given our experimental analysis showcasing how effective our simple approach is regarding expert burden management in interactive settings compared to previous approaches, we expect that this paper  brings a new valuable method to the community.

We would like to point out that we will provide an **open-source implementation of RND-DAgger** and our 4 considered baselines (DAgger, Lazy-Dagger, Ensemble-DAgger and HG-DAgger) **along with all environments**, including interactive training and evaluation procedures. The code is already available in supplementary material, and will be uploaded to a github codebase upon acceptance. We hope this will catalyze further research in this field.

---

> ### Author Response · Authors · 2024-11-22
> **Answer to all reviewers (2/3)**
>
> # On the impact of Minimal Expert Time
>
> Multiple reviewers (R-r6sZ,R-dV7L,R-Qs6v) asked for more information and/or experiments regarding the impact of the Minimal Expert Time component of the RND-DAgger approach. We thank them for pointing out this lack of clarity in the submitted manuscript. To overcome this problem, we both included results present in appendix C in the main paper together with conducting additional experiments to better understand the impact of this parameter.
>
> **We do ablate the Minimal Expert Time component of our approach in Appendix C**. Results show that removing MET slightly decreases task performance but drastically increases context switching (i.e. expert burden). In the updated manuscript, we will clearly refer to and briefly discuss the ablation study available in Appendix C.
>
> ## Additional Experiments
>
> Related to the impact of Minimal Expert Time, **R-r6sZ** wondered to what extent would baseline methods benefit from using the same approach, which is also related to **R-dV7L** wondering whether RND-DAgger is better than Lazy-DAgger just because it is using this Minimal Expert Time mechanism.
>
> Following these concerns, we are happy to share **additional experimental results**. More precisely, we implemented the **Minimal Expert Time (MET)** component into the Ensemble-DAgger and Lazy-DAgger baselines. For a fair comparison, as done initially for RND-DAgger, we conducted a **hyperparameter search** over a range of possible sizes W of this MET (see Table 1 of Appendix A). In the following table, we showcase these new augmented baselines **on all of our test environments** (8 seeds, mean perf. and 95% confidence intervals):
>
> | Method                   	| RC Task Performance| HC Task Performance | Maze Task Performance| RC Context Switch | HC Context Switch | Maze Context Switch |
> |------------------------------|-----------------------------|-----------------------------|-------------------------------|----------------------------|----------------------------|-----------------------------|
> | **Lazy-DAgger + MET**	| 0.943 $\pm$ 0.011      	| 2374 $\pm$ 325         	| 0.575 $\pm$ 0.101        	| 759 $\pm$ 39          	| 7 $\pm$ 0             	| 876 $\pm$ 71           	|
> | **Lazy-DAgger**          	| 0.939 $\pm$ 0.025      	| 2314 $\pm$ 278         	| 0.590 $\pm$ 0.057        	| 3437 $\pm$ 89         	| 312 $\pm$ 45          	| 1238 $\pm$ 79          	|
> | **Ensemble-DAgger + MET**| 0.935 $\pm$ 0.019      	| 2502 $\pm$ 228         	| 0.707 $\pm$ 0.023        	| 77 $\pm$ 4            	| 315 $\pm$ 84          	| 2034 $\pm$ 79          	|
> | **Ensemble-DAgger**      	| 0.952 $\pm$ 0.018      	| 2489 $\pm$ 108         	| 0.626 $\pm$ 0.045        	| 2785 $\pm$ 41         	| 1452 $\pm$ 134        	| 2871 $\pm$ 94          	|
> | **RND-DAgger**           	| 0.944 $\pm$ 0.014      	| 2490 $\pm$ 160         	| 0.717 $\pm$ 0.018        	| 368 $\pm$ 11          	| 708 $\pm$ 130         	| 1214 $\pm$ 25          	|
> | **RND-DAgger w/o MET**	| -       	| 2432 $\pm$ 175         	| -                         	| -         	| 1985 $\pm$ 215        	| -                      	|
>
>
>
>
>
> Regarding **Lazy-DAgger + MET**, for task performance, RND-DAgger is comparable to this baseline on Race Car and Half-Cheetah, and it statistically significantly outperforms Lazy-DAgger + MET in the goal-conditioned maze environment. RND-Dagger is however more expert-data sample efficient than LazyDagger + MET, as it reaches **significantly higher performances along training**: see figure 13b of updated appendix C in the updated manuscript. Regarding the expert burden (number of context switching), Lazy-DAgger significantly underperforms w.r.t. our approach on Race Car. On Half-Cheetah, there is no context switching for Lazy-DAgger due to the MET mechanism poorly combining with their hysteresis band criterion: when expert intervention is triggered, control is never given back to the agent.
>
> Regarding **Ensemble-DAgger + MET**, RND-DAgger is comparable in task performance to this approach on all environments. Regarding context switches for the expert, RND-DAgger underperforms compared to Ensemble-DAgger + MET which greatly benefits from this augmentation. That being said, we would like to recall that the **Ensemble-DAgger method requires that the expert provide the expert action along the entire evaluation phase (during but also outside actual expert interventions)**.
>
> In conclusion, adding the Minimal Expert Time mechanism does improve previous baselines regarding their required number of context switching (although it can lead to edge cases with Lazy-DAgger). However the combination of MET with a RND-based criterion – **RND-DAgger** – **is further confirmed as striking the best compromise between task performance, expert sample efficiency and the number of context switching without requiring constant expert awareness** (unlike for Ensemble-DAgger).

---

> ### Author Response · Authors · 2024-11-22
> **Answer to all reviewers (3/3)**
>
> # Considered baselines
>
> Multiple reviewers (R-USwa, R-r6sZ and R-Qs6v) pointed out a potential lack of enough baseline methods and suggested multiple works from the literature. We thank the reviewers for pointing out these interesting works. While we agree we should position w.r.t. these works, we respectfully disagree that they need to be included in our experimental analysis. We already compare to a diversity of established interactive imitation learning approaches: DAgger (vanilla procedure), EnsembleDAgger (ensemble-based expert action discrepancy criterion), Lazy-DAgger (error between action and predicted expert action) and HG-DAgger (human expert criterion). In the following paragraphs, we briefly discuss and position RND-DAgger to the suggested literature.
>
>  > Luo, Jianlan, et al. "RLIF: Interactive Imitation Learning as Reinforcement Learning." arXiv
>
> RLIF is an interesting method presenting a new approach to leverage expert interventions. Instead of learning behaviors from collected expert intervention data, RLIF learns through a reinforcement learning signal to avoid expert interventions. RLIF does not present a new approach for time-efficient expert control-taking: they consider a HG-Dagger setup, where the experimenter observes the interaction and takes control when a problem is observed.
> As such it is an interesting but complementary method, orthogonal to our work. In future work one could study how to combine efficient control-taking mechanisms like RND-Dagger with RLIF.
>
>
> > S. Dass et al. Pato: Policy-assisted teleoperation for scalable robot data collection. arXiv
>
> PATO does present a method allowing a learning agent to trigger expert control based on uncertainty. Their learning agent is a hierarchical goal-conditioned agent composed of a subgoal predictor network (a latent and goal-conditioned VAE) and an ensemble of subgoal reacher networks. Their uncertainty measure is based on
> * A) Policy uncertainty, which is similar to EnsembleDagger, i.e. using the disagreement between their ensemble of subgoal reacher policies.
> * B) Task uncertainty, measured as the variance between sampled subgoals along the agent trajectory.
>
> While this approach could have been considered, we argue that EnsembleDagger is a more informative comparison. PATO assumes a complex hierarchical setup with learned VAEs and an ensemble of low-level policies to tackle goal-conditioned setups. Our approach is more general, and as such we evaluate RND-DAgger in both classical (Half-Cheetah, RaceCar) and goal-conditioned (Maze) environments. As a final note, featuring a comparison with PATO would be complex as the authors did not release their code https://github.com/clvrai/pato/blob/main/README.md.
>
> > H. Liu et al. Model-based runtime monitoring with interactive imitation learning. ICRA, 2024
>
> In this work, the authors present a model-based approach to trigger expert intervention based on predicted failures. To do so, authors learn a VAE-based forward model of the environment, used to produce imaginary rollouts (within a learned latent space). The authors also train a failure (latent) state classifier to assess the quality of terminal planned (latent) states.
>
> After generating N rollouts of length L from current state s_t onward, their expert intervention triggering is based on two measures:
>
> A) Out-of-distribution detection, for which they compute the average Euclidean distance of their N terminal (planned) states w.r.t. closest states of the training data (nearest neighbor algorithm).
>
> B) Failure detection, i.e. if not triggered by the OOD threshold, intervention can also be triggered by exceeding a predefined average “failure state” percentage measured by using the failure classifier over terminal (planned) states.
>
> This approach considers an interesting but expensive planning-based strategy to trigger interventions, which relies on computing multiple rollouts and performing nearest-neighbor queries over the (growing) in-distribution training data. Besides, this method is specifically designed for predictive expert triggering, a direction that is interesting as pointed out by R-USwa but that we leave for future work.
> Most importantly, as for the previously discussed work, the authors did not release their codebase, which makes it hard to feature a comparative analysis w.r.t. RND-DAgger.
>
> > S.-F. Chen et al. Diffusion model-augmented behavioral cloning. ICML, 2024.
> > P. Florence et al. Implicit behavioral cloning. CoRL, 2022.
>
> Both works present improved Imitation Learning approaches compared to standard approaches such as Behavioral Cloning. These works do not address the interactive imitation learning problem, i.e. how to better leverage expert interventions to accelerate data collection and improve performances. As such they are orthogonal works: future work could study how to combine interactive imitation learning frameworks with improved imitation learning.

---

> ### Author Response · Authors · 2024-11-26
> **Answer about experiments**
>
> We thank the reviewers for taking the time to read our answers and to provide their concerns about our new results.
> As we agree with your interpretations of the updated Table 1, and our new experiments on the MET criterion, we would like to discuss further one of the key disadvantages of the  Ensemble-DAgger baseline. A crucial aspect that was not sufficiently emphasized in our previous responses is the total expert time required. While focusing solely on task performance and context switches provides important insights, it does not tell the complete story about expert burden. We conducted an additional analysis on the total expert time, as we did in the fig 8b) on human experiments. This table is now added in the Appendix as Table 5.
>
> | Method                         	| RC Task Performance | HC Task Performance | Maze Task Performance | RC Total Expert Time     | HC Total Expert Time             	| Maze Total Expert Time         	|
> |-----------------------------|-----------------------------|-----------------------------|-------------------------------|--------------------------------------|--------------------------------------|---------------------------------------|
> | **RND-DAgger**                     	| 0.944 $\pm$ 0.014                 	| 2490 $\pm$ 160           | 0.717 $\pm$ 0.018                    	| 16664 $\pm$ 73 (27.8 $\pm$ 0.1 mins) | 11968 $\pm$ 107 (6.60 $\pm$ 0.06 mins) | 16173 $\pm$ 44 (27.0 $\pm$ 0.1 mins)  |
> | **Ensemble-DAgger**          	| 0.952 $\pm$ 0.018                 	| 2489 $\pm$ 108           | 0.626 $\pm$ 0.045                    	| 31394 $\pm$ 506 (52.3 $\pm$ 0.8 mins) | 21535 $\pm$ 774 (11.8 $\pm$ 0.4 mins) | 51434 $\pm$ 1507 (85.7 $\pm$ 2.5 mins) |
> | **Ensemble-DAgger + MET**   | 0.935 $\pm$ 0.019              	| 2502 $\pm$ 228           | 0.707 $\pm$ 0.023                    	| 18216 $\pm$ 165 (30.4 $\pm$ 0.3 mins) | 13474 $\pm$ 631 (7.40 $\pm$ 0.35 mins) | 48125 $\pm$ 900 (80.2 $\pm$ 1.5 mins) |
>
> As we see, **Ensemble-DAgger is highly time consuming for the expert without statistically outperforming RND-Dagger**. And even with the addition of our MET mechanism on that baseline, we are still more efficient.
> Moreover, we would also like to insist on the human expert scenario (especially in a fast-paced environment like RaceCar).The Ensemble-DAggert baseline, even its improved MET variant, **requires the expert to input actions at the same time as the agent to be able to compute the discrepancy measure, which is not natural in practice**.
>
> LazyDAgger indeed solves that last issue, but at a task performance cost. We kindly remind the reviewers that on the only task on which LazyDAgger has less context switching than RND-Dagger (HalfCheetah), our method is more efficient in the early stages of the experiments (see Fig 6b) and 13b) of the revised paper for the MET variant). In other words, the improvements in expert burden for LazyDAgger comes at a cost in task performances.
>
> Given this analysis, we can summarize our key contributions highlighted by our experiments:
>
> 1. A simple yet efficient method for interactive imitation learning that significantly reduces expert time requirements while maintaining or improving performance
>
> 2. An open-source benchmark implementation of multiple interactive imitation learning approaches
>
> 3. Empirical demonstration of improved sample efficiency and reduced expert burden, particularly in goal-conditioned tasks
>
> 4. A practical approach that does not require constant expert monitoring, making it more feasible for real-world applications
>
> In addition, we have updated the paper to include a discussion about the limitations of the approach (see updated section 5 of the manuscript)

---

### Meta-Review · Area_Chair_p1iG · 2024-12-20

**Metareview:**

(a) Summary: This paper presents Random Network Distillation DAgger (RND-DAgger), an active imitation learning method that leverages RND to define out-of-distribution states, enabling selective expert intervention and minimizing the frequency of transitions between human experts and learning agents through a minimal demonstration time mechanism.
(b) Strengths: The paper is generally well-written and easy to follow. The proposed approach seems interesting and reasonable. The experimental results seem to support the authors' claims.
(c) Weaknesses: The reviewers pointed out a few major concerns and questions. The novelty is somewhat incremental. Some comparisons with baselines are missing.
(d) The authors have addressed the majority of the concerns from reviewers. All reviewers gave a final rating of above borderline.

**Additional Comments On Reviewer Discussion:**

Some of the reviewers replied to the authors' rebuttal and appreciated the clarification and refinement of the paper.

---

### Decision · Program_Chairs · 2025-01-22

Accept (Poster)